# Caveolar communication with xenobiotic-stalled ribosomes compromises gut barrier integrity

Seong-Hwan Park[1,2,3], Juil Kim[1,3] & Yuseok Moon [1✉]

In response to internal and external insults, the intestinal lining undergoes various types of epithelial adaptation or pathologic distress via stress-responsive eIF2α kinase signaling and subsequent cellular reprogramming. As a vital platform for growth factor-linked adaptive signaling, caveolae were evaluated for epithelial modulation of the insulted gut. Patients under ulcerative insult displayed enhanced expression of caveolin-1, the main structural component of caveolae, which was positively associated with expression of protein kinase R (PKR), the ribosomal stress-responsive eIF2α kinase. PKR-linked biological responses were simulated in experimental gut models of ribosome-inactivating stress using mice and *Caenorhabditis elegans*. Caveolar activation counteracted the expression of wound-protective epidermal growth factor receptor (EGFR) and its target genes, such as chemokines that were pivotal for epithelial integrity in the ribosome-inactivated gut. Mechanistic findings regarding ribosomal inactivation-associated disorders in the gut barrier provide crucial molecular evidence for detrimental caveolar actions against EGFR-mediated epithelial protection in patients with IBD.

[1] Laboratory of Mucosal Exposome and Biomodulation, Department of Biomedical Sciences and Biomedical Research Institute, Pusan National University, Yangsan 50612, Korea. [2] Present address: Personalized Genomic Medicine Research Center, Korea Research Institute of Bioscience and Biotechnology, Daejeon 34141, Korea. [3] These authors contributed equally: Seong-Hwan Park, Juil Kim. ✉email: moon@pnu.edu

nflammatory bowel disease (IBD) is strongly influenced by crosstalk between genetic and environmental etiologies, thus leading to the disruption of epithelial integrity and subsequent inappropriate mucosal immune responses[1,2]. In the gut luminal environment, specialized membrane microdomains (lipid rafts) of enterocytes are vital platforms for regulating biological responses via the regulation of plasma and endosomal membrane receptor-associated signaling transduction. Caveolae, a special type of lipid raft, are small (50–100 nm) invaginations in the plasma membrane that provide a platform for signaling molecule assembly, thus influencing membrane fluidity and membrane protein trafficking, and regulating receptor trafficking[3]. In particular, multiple signaling receptors migrate to the caveolae to facilitate signal transduction for sensing changes in the external environment and initiating events that alter cellular physiology, such as cell-to-cell communication and the activation of effector molecules for defending against external insults[4,5]. In response to ulcerative injuries, cells undergo various reprogramming processes for tissue repair and restitution that are regulated by signals from growth factors and chemokines. In particular, epidermal growth factor (EGF) plays pivotal roles in epithelial restitution and wound healing in ulcers and disconnected barrier lesions in response to luminal stress[6]. EGF receptor (EGFR)-activated epithelial modulation is thus crucial for protection against luminal stressors, including dietary xenobiotic insults and mucosal infection.

In response to diverse internal or external stressors, eukaryotic cells activate a common adaptive pathway, known as the integrated stress response (ISR), to restore cellular integrity. The core biochemical event in ISR is the phosphorylation of eukaryotic translation initiation factor 2 alpha (eIF2α) by the eIF2α kinase family, leading to global translational arrest and the induction of specific stress-responsive genes to achieve biological homeostasis[7,8]. The alpha subunit of eIF2 is targeted by four different stress-related mammalian protein kinases, including double-stranded RNA-dependent protein kinase R (PKR), RNA-dependent protein kinase-like ER kinase (PERK), eIF2α kinase general control non-repressed 2 (GCN2), and heme-regulated eIF2α kinase (HRI)[7,8]. Upon exposure to internal or external stressors, including viral infection, ribosomes stand sentinel. In particular, stress-driven ribosomal stalling triggers eIF2α-mediated global translational inhibition via PKR, which is a primary biochemical pathway in ISR[7,9–11]. In addition to ribosomal inactivation in response to viral infection or specific translational inhibitors, other internal or external stressors, such as specific oxidative and ER stress, growth factor deprivation, or bacterial infection, can induce ribosomal stress, leading to ISR via PKR activation[7,12]. Ribosomal inactivation is also a potent trigger of epithelial inflammatory and malignant disorders, including IBD[13–15]. Mechanistically, despite inhibition or cleavage of 28S ribosomal RNA at the peptidyl transferase center during gene translation[16–19], ribosomal inactivation may result in the expression of some crucial genes involved in cellular homeostasis and pathogenic processes[17,20]. In particular, the cytokine and growth factor profiles are highly perturbed in the epithelia under ribosomal stress, leading to mucosal inflammatory injuries and potentially chronic outcomes, such as tumor formation. However, little information is available regarding intracellular communications between the stalled ribosome and cell surface platforms in response to ribosome-derived stress, which may initiate adverse outcomes or restore biological homeostasis.

In the present study, gut barrier distress was associated with ISRs as adaptive pathways in human intestinal inflammatory disease. In particular, PKR-linked stress responses were simulated in experimental models of ribosomal inactivation. This study assessed whether stress-driven ribosomal inactivation may alter communications through actions by caveolae as a key platform of stress response signaling for growth factor receptor-mediated homeostasis. The mechanistic investigation in view of stress-driven ribosomal inactivation will provide insights into gastrointestinal modulation of caveolae-linked events in epithelial disorders, such as IBD or colitis-associated tumors.

## Results

**Caveolin-1 expression is linked to ISRs in patients with IBD**. As a key component of caveolae, caveolin-1 (Cav-1) expression was compared between healthy and IBD patient groups based on clinical genomic datasets (Fig. 1a). In two datasets, Cav-1 expression was higher in patients with ulcerative colitis (UC) and Crohn's disease (CD) than that of each control. Moreover, Cav-1 protein levels in colonic biopsies were higher in patients with IBD than in the healthy controls (Fig. 1b). Consistent with the observations in human IBD, mice with chemical-induced UC also displayed elevated Cav-1 transcriptional (Fig. 1c) and protein levels (Fig. 1d, e) in the gut compared to controls. Based on the assumption that the changes in Cav-1 as the signaling platform may reflect stress responses to chronic luminal insults in patients with IBD, four different global stress-related mammalian eIF2α kinases, including EIF2AK1 (HRI), EIF2AK2 (PKR), EIF2AK3 (PERK), and EIF2AK4 (GCN2), were evaluated in both healthy and IBD patient groups based on a clinical genomic dataset (Fig. 1f). Among these, PKR was notably elevated in patients with UC and CD, including colon-only CD (cCD) and ileo-colonic CD (iCD), when compared to the control group. Moreover, disease-active patients displayed enhanced PKR and Cav-1 levels compared to normal controls or disease-inactive IBD groups (Fig. 1g). In terms of signaling pathways, IBD patients with high PKR expression tended to show increased levels of PKR-activated stress-responsive transcription factors, such as activating transcription factor 4 (ATF4) and C/EBP homologous protein (CHOP) (Fig. 1h). Finally, the same clinical datasets demonstrated that IBD patients with high PKR expression also displayed enhanced Cav-1 expression (Fig. 1i), indicating a positive association between Cav-1 and PKR-linked stress responses.

**Cav-1 regulates expression and nuclear localization of EGFR.** Next, PKR-linked stress responses were simulated in the experimental models of ribosomal stress via direct activation of PKR[10,19]. We assessed the effects of PKR activation on Cav-1 expression in the ribosomal stress-insulted murine gut and human cell models. Upon exposure to internal or external stressors, ribosomes stand sentinel. Stress-driven ribosomal stalling triggers eIF2α-mediated global translational inhibition via PKR, which was mimicked by exposure to chemical ribosome-inactivating stressors (RIS). RIS-exposed mice displayed elevated intestinal Cav-1 expression (Fig. 2a, b). Consistent with clinical transcriptomic analysis in patients with IBD (Fig. 1h), RIS-exposed cells showed initially increased levels of PKR-activated stress-responsive transcription factors, such as ATF3, ATF4, and CHOP, while expression of ER chaperone glucose-regulated protein 78 (GRP78) was suppressed (Fig. 2c). Moreover, caveolar components such as cavin-1 and Cav-1 displayed upregulated expression. Compared with early induction of ATF3, CHOP, and cavin-1, late induction was observed for Cav-1, including alpha and beta isoforms, in response to ribosomal inactivation (Fig. 2c, d). The alpha isoform was more responsive than beta isoforms to ribosomal stress. Although there were differences in their degrees of action, RIS-1 and RIS-2 showed similar patterns of gene regulation for PKR-associated integrated stress responses (ISR). However, since RIS-2 had more remarkable effects on Cav-1 induction than RIS-1 (Fig. 2c, d), it was used as a representative modulator of caveolin-linked signaling under ribosomal

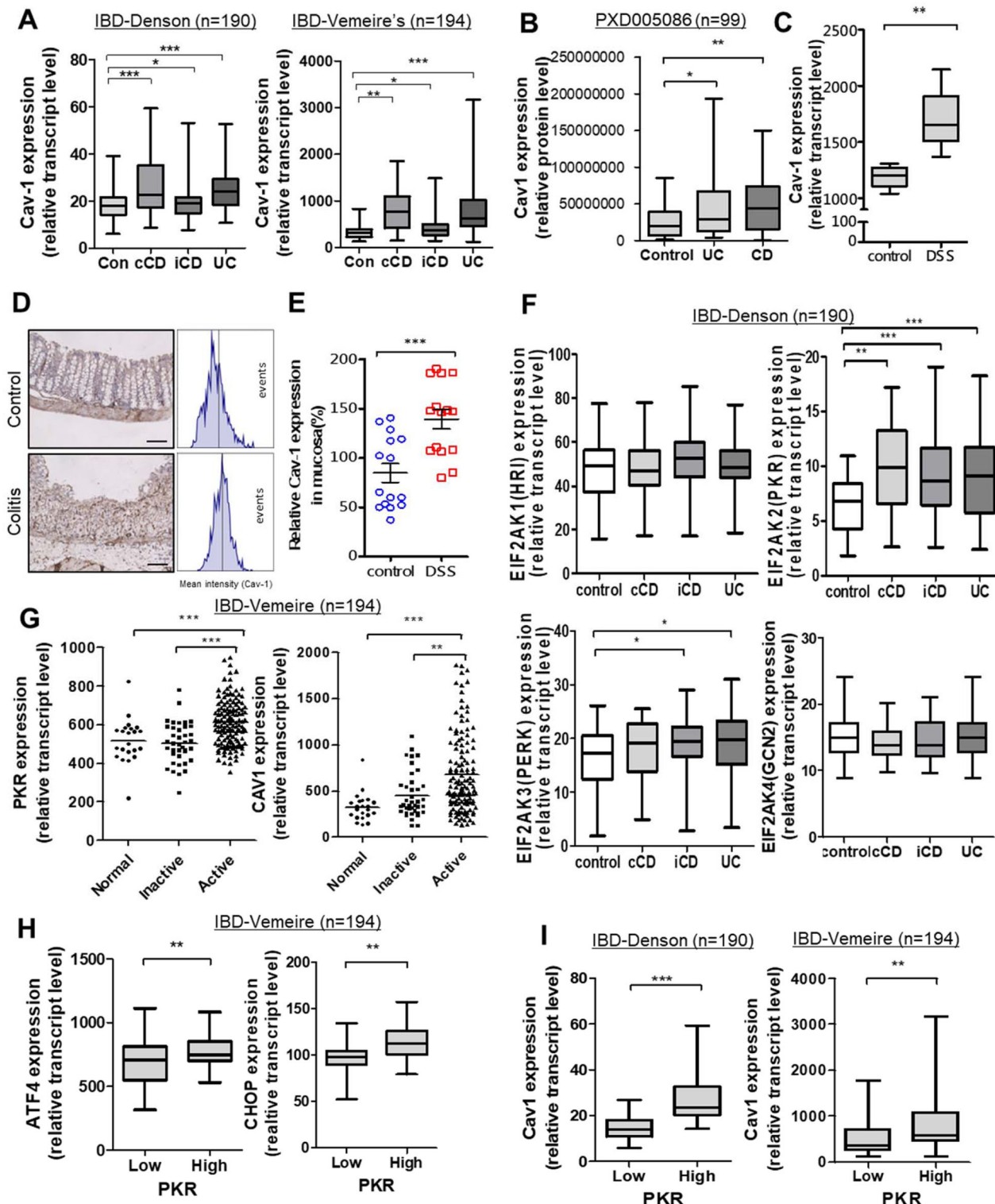

inactivation stress in this study. In terms of gene regulation, ATF3 is a key transcription factor for stress-responsive genes during translational arrest[21] and was tested for its effects on subsequent Cav-1 expression. We found that ATF3 was positively involved in inducing Cav-1 expression under ribosomal inactivation stress (Fig. 2e and Supplementary Fig. 2). Moreover, cellular ribosomal inactivation enhanced Cav-1 protein levels, which were maintained for 48 h in human intestinal epithelial cells (HCT-8) (Supplementary Fig. 1A). Other enterocyte-derived cell lines (HT-29 and SW-480) also showed enhanced Cav-1 expression in

response to ribosomal inactivation (Supplementary Fig. 1B). HCT-8 cells are widely used as a human intestinal epithelial cell model for inflammatory and infectious diseases[22,23]. In particular, the source of the HCT-8 cell line in the ileocecum region of the small intestine is particularly susceptible to ribosomal inactivation-associated ISR[24,25]. In addition to the effects on total Cav-1 protein levels, microscopic analysis demonstrated that ribosome-inactivated HCT-8 cells displayed clustering of Cav-1 protein at the lipid raft caveolae (Fig. 2f, g). Thus, ribosomal inactivation was investigated to determine whether it also alters

**Fig. 1 Involvement of ISR and Cav-1 in IBD. a** Intestinal Cav-1 expression was compared in patients with different IBD types in datasets gse117993 (*Denson's*, $n = 190$) and gse75214 (*Vemeire's*, $n = 194$). **b** Colonic Cav-1 protein expression was compared in patients with different IBD types in a proteomic dataset (PXD005086, $n = 99$). **c** Intestinal Cav-1 expression was compared in DSS-exposed mice using the dataset (gse22307, $n = 6$-7). **a**–**c** Results are shown as box-and-whisker plots (min to max) and asterisks (*) indicate significant differences from the control group (*$p < 0.05$, **$p < 0.01$, ***$p < 0.001$ using two-tailed unpaired Student's $t$ test). **d**, **e** Cav-1 expression was assessed in colons from 8-week-old female C57BL/6 mice treated with vehicle or 3% (w/v) DSS ($n = 14$-15, each group). Microscopic analysis was conducted at 200× magnification; scale bar(s): 100 μm. **d** Relative Cav-1 density was measured using the HistoQuest tissue analysis software (each histogram represents events with increasing DAB levels, as denoted in the method in detail). **e** Quantitative comparisons are shown in the graph. Asterisks (***) indicate a significant difference from the control group ($p < 0.001$ using two-tailed unpaired Student's $t$ test). **f**, **g** Intestinal expression of eIF2a kinases (EIF2AK1, EIF2AK2, EIF2AK3, or EIF2AK4) or Cav-1 was compared in patients with different IBD types from datasets gse117993 (**e**, *Denson's*, $n = 190$) and gse75214 (**f**, *Vemeire's*, $n = 194$). UC, ulcerative colitis; cCD, colon-only CD; iCD, ileo-colonic CD. Results are shown as box-and-whisker plots (min to max). Asterisks (*) indicate significant differences between groups (*$p < 0.05$, **$p < 0.01$, ***$p < 0.001$ using two-tailed unpaired Student's $t$ test). **h**, **i** Expression levels of PKR target genes were assessed in patients with IBD (gse75214 [$n = 194$] or gse117993 [$n = 190$]). Based on PKR levels, we chose the 50 highest and 50 lowest samples, which were further compared for expression of ISR-linked genes (ATF4 and CHOP) (**h**), or Cav-1 (**i**). Results are shown as box-and-whisker plots (min to max). Asterisks (*) indicate significant differences from the low-expression group (**$p < 0.01$, ***$p < 0.001$ using two-tailed unpaired Student's $t$ test).

caveolae-modulated signaling receptors and their localization in human intestinal cells. As a potent caveolae-regulated signaling receptor, EGFR was evaluated in the tissues of patients with IBD. Compared to control levels, these patients showed decreased EGFR expression in the gut, with UC patients showing particularly notable reductions (Fig. 2h). Moreover, IBD patients with high Cav-1 levels tended to display attenuated EGFR expression in the gut compared to the low-expression group (Fig. 2i). Further analysis of the correlation between Cav-1 and EGFR levels confirmed that EGFR expression was inversely related with Cav-1 levels in mucosal biopsies from patients with IBD (Supplementary Fig. 1C). In terms of ISR, high-PKR patients showed attenuated intestinal EGFR expression (Fig. 2j), indicating potent negative regulation of EGFR during PKR activation. Furthermore, the effects of Cav-1 on EGFR-associated activities were evaluated in PKR-activated human intestinal cells. PKR-activating RIS transiently induced total EGFR, and the maximum induction level was elevated in Cav-1-deficient cells (Supplementary Fig. 1D). At maximal EGFR induction (1 h), the cellular fraction analyses of proteins showed that EGFR was induced, and some quantities of induced EGFR moved into the nuclear regions in response to PKR activation, particularly in Cav-1-deficient cells (Fig. 2k and Supplementary Fig. 2). The main cellular feature of ribosomal inactivation was the induction and subsequent partial nuclear translocation of EGFR; however, this was counteracted by Cav-1-associated activity in the intestinal epithelial cells. This cellular evidence provided by the ribosomal inactivation model using human cells also suggests an inverse relationship between Cav-1 and EGFR. Cav-1-mediated regulation of EGFR expression and its biological activities were further assessed in the following experiments.

**Cav-1 counteracts EGFR-mediated maintenance of gut barrier.**
Based on the hypothesis that Cav-1 may interfere with EGFR-linked activities in the ribosome-inactivated gut, we investigated a model of *Caenorhabditis elegans*, which is particularly dependent on the gut epithelial layer for defense because it lacks any "professional" leukocytes, such as macrophages and lymphocytes, to defend against pathogens. Ribosomal inactivation deteriorated the epithelial lining of the worm and reduced the gut cell number, but these suppressive effects were attenuated by Cav-1 mutation. In contrast, the ribosomal inactivation-induced epithelial injuries were not restored by EGFR (Let-23) mutation (Fig. 3a, b). EGFR-linked barrier protection was verified by visualizing luminal contents in nematodes using a non-absorbable blue food dye. While most dye could be detected within the nematode gut lumen of the wild-type control, the ribosome-inactivated worms displayed staining in the body cavity in addition to the gut lumen,

indicating epithelial barrier disruption (Fig. 3c, d). However, Cav-1-mutated animals had reduced body cavity staining, supporting the barrier-damaging roles of caveolae in response to ribosomal inactivation. In contrast, Cav-1-regulated EGFR was positively involved in protecting against severe epithelial barrier leakage due to ribosomal inactivation in *C. elegans*. As the gut barrier-based immunity of *C. elegans* is crucial for defending against microbial invasion, the ribosome-insulted gut allowed bacterial overgrowth (Fig. 3e, f). In contrast, caveolin mutation attenuated the gut microbial population, whereas the gut defense against bacteria under ribosomal inactivation was aggravated by EGFR inhibition.

Consistent with the EGFR-associated beneficial activities in the *C. elegans* gut model, ribosomal inactivation caused ulcerative injuries in the small intestine of mice, which were aggravated by EGFR inhibition (Fig. 3g). EGFR-mediated protection counteracted the ribosomal inactivation-induced pathologic severities, including inflammation, crypt loss, and edema (Fig. 3h). Closer observation of the murine intestinal epithelial barrier by *F*-actin staining indicated that ribosomal inactivation altered the apical barrier integrity (Fig. 3i, j). Notably, EGFR inhibition severely disrupted *F*-actin-based apical projections in the mouse gut epithelial lining, whereas more *F*-actin molecules spread into the basolateral and basement areas of the enterocytes. Taken together, this evidence indicates that EGFR mediates protection of epithelial barrier integrity in the gut during ribosomal inactivation.

**EGFR-induced chemokines are crucial to gut barrier integrity.**
Pathway analysis of human clinical samples demonstrated that inflammation-associated pathways, particularly those involved in chemokine-linked signaling, were highly correlated with EGFR expression in the intestinal tissues of patients with IBD (Fig. 4a). Among EGFR-correlated genes in IBD patients, C–C chemokine receptor type 2 displayed the highest correlation. Interleukin 8 (IL-8) and chemokine (C–X–C motif) ligand 1 (CXCL-1) bind to C–C chemokine receptor type 2 as a primary receptor involved in their biological activities. We further assessed the murine insult model to determine if EGFR is a crucial signaling module for chemokine production in ribosome-inactivated enterocytes. Ribosomal inactivation also enhanced the expression of intestinal EGFR (Fig. 4b, c) and Cav-1 (Fig. 2a, b) in the murine model. Moreover, blockage of EGFR signals using genetic suppression by shRNA or chemical inhibitors suppressed IL-8 secretion in ribosome-inactivated human intestinal cells in vitro (Fig. 4d and Supplementary Fig. 1E). In addition to the effects on chemokine secretion, the mRNA expression of IL-8 and CXCL-1 was also attenuated by blocking EGFR-linked pathways in the ribosome-inactivated human intestinal cells (Supplementary Fig. 1F),

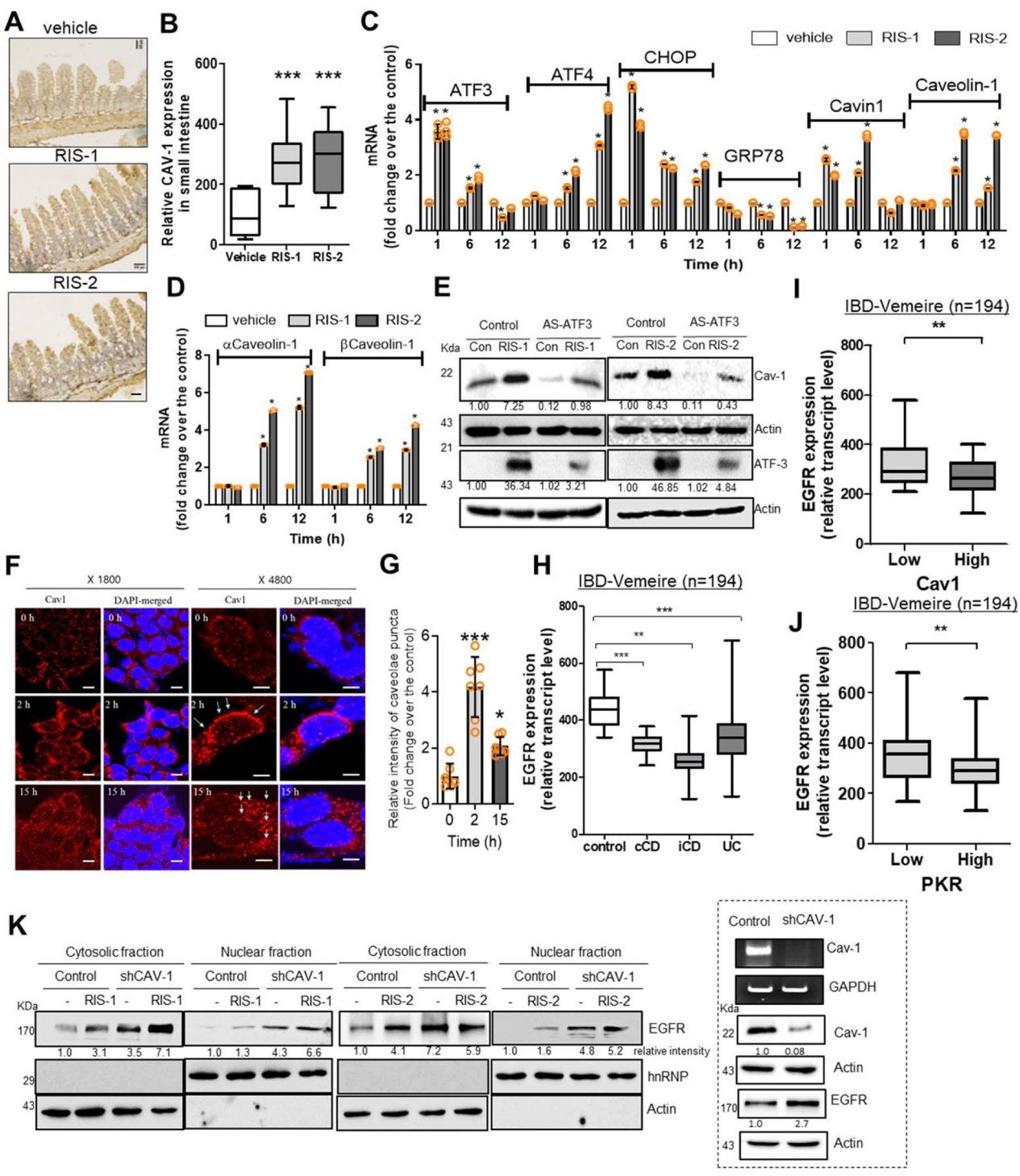

indicating that EGFR signals were involved in chemokine induction by ribosomal inactivation. As demonstrated in Fig. 2, Cav-1 negatively regulated EGFR expression in the ribosome-inactivated enterocytes; thus, it can be assumed that Cav-1 also downregulated EGFR-mediated chemokine production in response to ribosomal inactivation. When suppressing Cav-1 expression via shRNA, IL-8 secretion was enhanced in ribosome-inactivated human intestinal cells (Fig. 4e). Moreover, the mRNA expression of human chemokines (IL-8 and CXCL-1) was also upregulated by genetic knockdown of Cav-1 expression, indicating that Cav-1 negatively regulated chemokine expression (Supplementary Fig. 1G). In contrast, overexpression of Cav-1

suppressed IL-8 and CXCL-1 mRNA expression in the ribosome-inactivated human intestinal cells (Supplementary Fig. 1H). Taken together, as a negative regulator of EGFR expression, Cav-1 suppressed chemokine overproduction in response to ribosomal inactivation.

Generally, inflammatory chemokines play key roles in mediating immune cell infiltration of inflammatory lesions. However, although epithelial chemokine production was down-regulated by EGFR inhibition, neutrophil recruitment was elevated rather than retarded by EGFR inhibition in the ribosome-inactivated murine gut (Fig. 4f, g). Macrophage infiltration was also elevated by EGFR suppression in the mouse

**Fig. 2 Cav-1 suppresses EGFR expression in enterocytes. a**, **b** Cav-1 expression was assessed in the ilea of 8-week-old male C57BL/6 mice treated with vehicle, 25 mg/kg RIS-1, or 25 mg/kg RIS-2 for 24 h (n = 12–19, each group). **a** Microscopic analysis was performed at 200× magnification; scale bar(s): 100 μm. **b** Quantitative comparisons are shown as box-and-whisker plots (min to max). **c**, **d** HCT-8 cells were treated with vehicle, 50 ng/mL RIS-1, or 500 ng/mL RIS-2 for the indicated time. The mRNA was measured using real-time RT-PCR. Results are shown as mean values ± SD (n = 3). (**e**) HCT-8 cells stably transfected with control or antisense ATF3 (AS-ATF3) expression vector were treated with vehicle, 50 ng/mL RIS-1, or 500 ng/mL RIS-2 for 15 h (Cav-1) or 3 h (ATF3). Cell lysates were subjected to western blot analysis. **f**, **g** HCT-8 cells were treated with vehicle or 500 ng/mL DON (RIS-2) for the indicated time. **f** Cav-1 and nuclear regions were visualized using anti-Cav-1 and DAPI, respectively. Microscopic analysis was performed at 1800× or 4800× magnification; scale bar(s): 20 or 10 μm, respectively. **g** Quantitative comparisons are shown in the graph. Results are shown as mean values ± SD (n = 7). **h** Intestinal EGFR expression was compared in patients with different IBD types (Vemeire's, gse75214, n = 194). Results are shown as box-and-whisker plots (min to max). **i,j** EGFR expression was assessed in patients with IBD based on the gene expression array of IBD patients (gse75214 [n = 194]). Based on Cav-1 (**i**) or PKR levels (**j**), we chose the 50 highest and 50 lowest samples, which were further compared for EGFR expression. Results are shown as box-and-whisker plots (min to max). **a**–**j** Asterisks (*) indicate significant differences between groups (*$p < 0.05$, **$p < 0.01$, and ***$p < 0.001$ using two-tailed unpaired Student's t test). **k** HCT-8 cells transfected with control or shCav-1 vector were treated with vehicle, 50 ng/mL RIS-1, or 500 ng/mL RIS-2 for 1 h. Cytosolic or nuclear fractions were subjected to western blot analysis.

intestine (Fig. 4h, i). Taken together, these findings suggest that EGFR-induced chemokines are no longer positively involved in inflammatory cell recruitment during ribosomal inactivation stress. Instead, increased inflammatory cell recruitment in the EGFR-inhibited gut can be explained by other causes. Thus, it was hypothesized that EGFR-induced chemokines play protective roles against pro-inflammatory stimuli in the gut barrier. Ribosomal inactivation decreased the mucosal barrier against luminal microbiota, which was further aggravated by EGFR inhibition in both the small and large intestines of mice (Fig. 4j, k), indicating that EGFR had protective roles against bacterial exposure in the mucosal layer. Moreover, in vitro analyses of the permeability of the human epithelial cell monolayer demonstrated that IL-8 was positively involved in EGFR-mediated barrier protection from ribosomal inactivation stress (Fig. 4l, m). These findings suggest that EGFR-induced chemokines are essential to epithelial restitution rather than inflammatory cell recruitment during ribosomal inactivation. Therefore, barrier disruption in the EGFR-inhibited gut increased exposure to luminal microbes and subsequent inflammatory stress.

**Caveolae control EGFR actions under ribosomal stress.** Ribosomal inactivation-linked ISR has been extensively associated with the cellular stress signaling response of mitogen-activated protein kinases (MAPKs);[17,20] thus, MAPKs were evaluated to determine whether they are involved in caveolae- and EGFR-linked signaling transduction in ribosome-inactivated intestinal cells. Loss- or gain-of-function analyses demonstrated that Cav-1 negatively regulated p38 MAPK activation via ribosomal inactivation in human intestinal cells (Fig. 5a, b and Supplementary Fig. 2). In contrast, Cav-1 suppression downregulated extracellular signal-regulated kinases (ERK) 1/2 and c-Jun N-terminal kinase (JNK). Among the three major MAP kinases, only p38 MAPK was negatively regulated as a downstream module of Cav-1-linked signaling in ribosome-inactivated human intestinal cells (Fig. 5a, b and Supplementary Fig. 2). Because Cav-1 counteracted EGFR signals in the present cell model (Fig. 2k), it was hypothesized that p38 MAPK mediates the suppression of EGFR-linked signaling by Cav-1. Notably, chemical inhibition of EGFR downregulated p38 MAPK signals in response to ribosomal inactivation (Fig. 5c and Supplementary Fig. 2), and genetic knockdown of human EGFR also attenuated the p38 signals in human intestinal cells (Fig. 5d and Supplementary Fig. 2). Consequently, Cav-1 counteracted EGFR-activated p38 MAPK, reducing chemokine secretion and expression in ribosome-inactivated human intestinal cells (Fig. 5e, f). In addition to the involvement of EGFR-mediated signaling in chemokine production, PKR, the ribosomal stress-responsive eIF2α kinase, was tested for its effects on p38 MAPK signaling and chemokine

expression. Subsequently, shRNA-mediated PKR suppression attenuated p38 activation (Fig. 5g and Supplementary Fig. 2) and chemokine induction by RIS (Fig. 5h). Moreover, PKR was involved in activating EGFR phosphorylation, leading to p38 MAPK activation (Fig. 5g and Supplementary Fig. 2). Altogether, these findings indicated that PKR is a positive upstream regulator of EGFR and p38 MAPK signaling and subsequent chemokine production under ribosomal inactivation stress in human intestinal epithelial cells.

Next, p38 MAPK-targeted transcription factors were assessed in the stressed cells. Our previous investigations demonstrated that early growth response 1 (Egr-1) is the most important transcription factor involved in chemokine induction in ribosome-inactivated intestinal epithelial cells[26,27]. Ribosome-inactivated human intestinal cells displayed enhanced nuclear expression of Egr-1 protein (Fig. 6a), which was notably dependent on p38 MAPK and ERK1/2 as upstream activators (Fig. 6b and Supplementary Fig. 2). Furthermore, Egr-1, as a key transcription factor of caveolae-associated signaling, was transcriptionally upregulated by p38 MAPK activation in ribosome-inactivated cells (Fig. 6c). Gain- or loss-of-function analyses demonstrated that Cav-1 downregulated Egr-1 expression in response to ribosomal inactivation (Fig. 6d, e and Supplementary Fig. 2). Cav-1-mediated negative regulation of Egr-1 expression was also confirmed based on its transcriptional activity (Fig. 6f). Because EGFR expression exhibited negative regulation by Cav-1 in patients with IBD (Fig. 2i) and human intestinal cells (Fig. 2k), Cav-1 and EGFR could have opposite effects on Egr-1 transcription. As expected, both genetic knockdown and chemical inhibition of EGFR reduced Egr-1 expression in ribosome-inactivated human intestinal cells (Fig. 6g, Supplementary Figs. 2 and 1I). Moreover, transcriptional activity of the Egr-1 promoter was attenuated by shRNA-mediated EGFR suppression (Supplementary Fig. 1J). Taken together, these findings indicate that EGFR-linked activation of p38 MAPK signaling mediated the transcriptional induction of Egr-1 and chemokines, all of which were counteracted by Cav-1 in ribosome-inactivated human intestinal cells.

**Cellular translocation of EGFR under ribosomal stress.** EGFR is a receptor tyrosine kinase (RTK) that transduces external signals into the internal environment, enabling the proper responses that maintain cell homeostasis. In addition to functioning as a surface receptor, EGFR can translocate into the nucleus to regulate gene transcription in response to particular stimuli[28]. In the present stress model using human intestinal HCT-8 cells, EGFR migrated into the nuclear region in response to ribosomal inactivation (Fig. 6h and Supplementary Fig. 2). To act as a nuclear regulator, EGFR can be internalized via clathrin-mediated

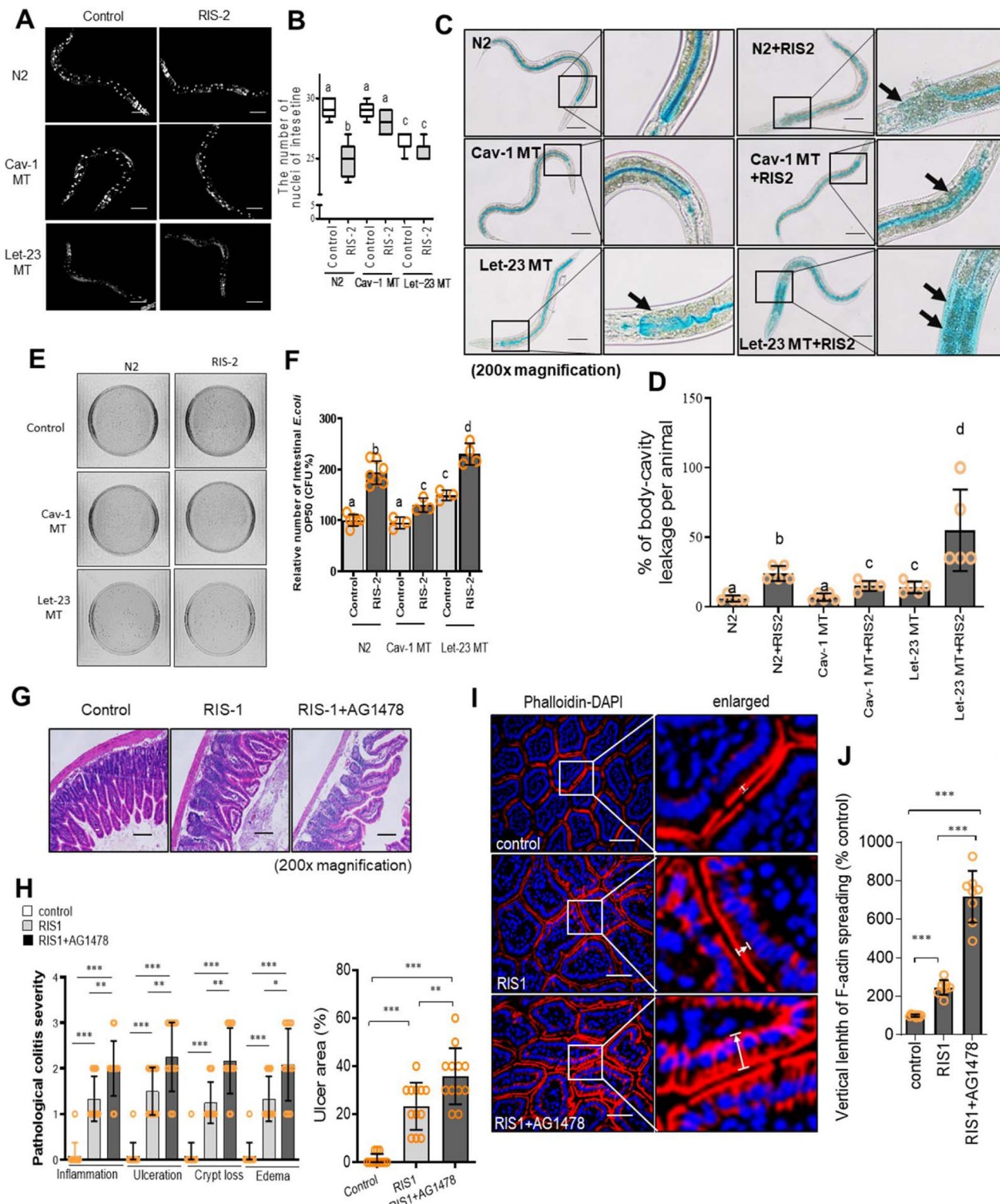

endocytosis (CME) or caveolae-dependent endocytosis;[29,30] these two pathways were analyzed by chemical blocking using mono-dansylcadaverine (MDC) or nystatin, respectively, which indicated that RIS-triggered transient nuclear translocation was mediated by CME (Fig. 6h and Supplementary Fig. 2). Moreover, inhibition of CME attenuated Egr-1 induction (Fig. 6i and Supplementary Fig. 2). Specifically, CME was positively involved in transcriptional activation of Egr-1 and chemokine promoters in the ribosome-inactivated human intestinal cells (Fig. 6j, k). Due to transcriptional modulation by Egr-1, the CME-dependent

pathway was positively involved in IL-8 production (Fig. 6l). In the present ribosomal inactivation model, p38 MAPK played key roles in transmitting signals from EGFR to induce transcription of Egr-1 and chemokine genes. Conversely, p38 MAPK was assessed for its involvement in nuclear translocation of EGFR protein as a downstream target of MAPK signaling in response to ribosomal inactivation. Notably, blocking of p38 MAPK down-regulated the nuclear translocation of EGFR in human intestinal cells (Fig. 6m and Supplementary Fig. 2). Confocal micro-scopy demonstrated partial nuclear translocation of EGFR

**Fig. 3 Effects of Cav-1 and EGFR on the gut barrier in the ribosome-inactivated gut. a**, **b** Worms (wild-type [N2], *caveolin-1* mutant [Cav-1 MT], and *EGFR* mutant [let-23 MT] nematodes, *n* = 6-9 animals per group) were treated with 500 ng/mL RIS-2 for 24 h, and then stained with DAPI (blue). **a** Fluorescence microscopic analysis was performed at 200× magnification; scale bar(s): 100 μm. **b** The graph indicates the number of intestinal nuclei per worm; results are shown as box-and-whisker plots (min to max). **c**, **d** Representative images of N2, Cat-1 MT, and let-23 MT nematodes after 3 h of staining with blue food dye and 24 h of RIS-2 treatment. **c** Black arrows indicate areas in which blue food dye has leaked from the gut lumen into the body cavity, causing the Smurf phenotype (200× magnification; scale bar(s): 100 μm). **d** The graph shows the percentage of body-cavity leakage per animal (*n* = 6 animals per group). Data represent the means ± SD (*n* = 5). **e**, **f** Detection (**e**) and quantitation (**f**) of viable intestinal *Escherichia coli* OP50 in N2, Cav-1 MT, and let-23 MT nematodes exposed to 500 ng/mL DON for 24 h. Data represent the means ± SD (*n* = 3–8). **a**–**f** Different letters over each bar or box represent significant differences between groups (*p* < 0.05 using one-way ANOVA with the Newman–Keuls post hoc test). **g**–**j** Eight-week-old male C57BL/6 mice received AG1478 (1 mg/mouse) intraperitoneally twice during 2 days before treatment with 25 mg/kg RIS-1 (*n* = 6–12 animals per group). Animals were sacrificed after RIS-1 treatment for 12 h. **g** Histological observation of hematoxylin and eosin (H&E)-stained small intestine sections. **h** pathological severity (200 ×; scale bar(s): 100 μm). **i** F-actin staining (red) and DAPI nuclear staining (blue). White arrows indicate F-actin localization altered from the apical distribution. Microscopic analysis was performed at 400 × magnification; scale bar(s): 50 μm. **j** Relative quantification of F-actin spread from the apical parts to basolateral or basal areas (%). **h**, **j** Asterisks (*) indicate significant differences between groups (**p* < 0.05, ***p* < 0.01, and ****p* < 0.001 using two-tailed unpaired Student's *t* test).

in ribosome-inactivated cells (Fig. 6n, o). In addition to influencing EGFR localization in cells, CME and p38 MAPK were positively involved in inducing total EGFR protein in response to ribosomal inactivation (Fig. 6n, o). Consistent with analyses of cytoplasmic and nuclear EGFR levels (Fig. 2k and Supplementary Fig. 1D), ribosomal inactivation mainly contributed to EGFR induction, after which some EGFR migrated into nuclei. All of these events (i.e., EGFR induction and nuclear translocation) were linked to CME in a p38 MAPK-dependent manner in response to ribosomal inactivation.

**Degradation of EGFR under ribosomal stress**. In terms of protein integrity, stress-induced EGFR protein can be susceptible to proteasome-induced degradation during intracellular trafficking for compartment protein targeting. In the present stress model using human intestinal HCT-8 cells, intestinal ribosomal inactivation triggered proteasome activation during ISR, which was positively regulated by Cav-1 protein (Fig. 7a, b). Cav-1 suppression reduced the proteasomal activity, which was reversed by Cav-1 overexpression. As a target of Cav-1-mediated proteasome activation, EGFR was assessed in ribosome-inactivated human intestinal cells. To facilitate the proteasome cascade, ubiquitin was introduced. EGFR ubiquitinylation was elevated by ubiquitin overexpression, resulting in EGFR protein suppression due to degradation (Fig. 7c and Supplementary Fig. 2). As a downstream EGFR-linked signal, p38 MAPK decreased with ubiquitinylation, leading to suppressed Egr-1 induction (Fig. 7c and Supplementary Fig. 2). Cav-1 was also tested to determine whether it regulates EGFR expression via proteasome-associated degradation. The results revealed that Cav-1-mediated EGFR reduction was notably attenuated by chemical-induced blocking of proteasome activation using MG132 in ribosome-inactivated human intestinal cells (Fig. 7d and Supplementary Fig. 2). Ultimately, ubiquitinylation promoted EGFR degradation in a proteasome-dependent manner, which contributed to down-regulating the expression of EGFR targets, including chemokines (Fig. 7e). Taken together, these findings indicate that Cav-1 counteracted the CME and nuclear action of EGFR and triggered proteasome-mediated degradation of EGFR protein followed by a subsequent reduction in downstream signaling in ribosome-inactivated human intestinal cells.

## Discussion
In the present study, we evaluated caveolar actions in the gut during stress-driven ribosomal inactivation, which is a potent molecular etiology of IBD. Ribosomal inactivation produced discordant oppositional activities between Cav-1 and EGFR-

linked signaling in the inflamed gut (Fig. 7f). Cav-1 was upregulated in lesions with reduced EGFR levels in patients, indicating that Cav-1 negatively mediated endogenous control of barrier-protective EGFR activity. In response to ulcerative injuries, the ribosomes stand sentinel, and stress-driven ribosomal inactivation regulates EGFR-linked responses via caveolar activation. Cav-1 is the principal structural component of caveolae, and it clusters in response to ribosomal inactivation. Ribosomal inactivation-associated ISR has been considered as a crucial mechanistic factor driving IBD progression in response to mucosal stressors. Functionally, ribosomal inactivation triggers CME and nuclear translocation of EGFR and the subsequent induction of target genes, such as chemokines, in human intestinal epithelial cells. In contrast, stress-associated caveolae formation interfered with nuclear translocation of EGFR and mediated EGFR protein degradation. Cav-1-mediated EGFR degradation can occur in the nucleus because the proteasome machinery functions also in the nuclear compartment[31]. In spite of proteasomal actions, ISR-responsive PKR activated p38, which was crucial for the cellular translocation and transcriptional actions of EGFR under ribosomal inactivation stress. Disruption of EGFR-mediated barrier protection allowed luminal microbes to access inner tissues, leading to pro-inflammatory stimulation in the gut under ribosomal stress. The molecular network of Cav-1 and EGFR signaling in the present study can account for biological adjustments in the insulted gut between reparative barrier maintenance and barrier disruption (Fig. 7f). Moreover, an assessment of stress responses verified the inverse relationship between Cav-1 and EGFR-linked signals in clinical datasets of patients with IBD.

An opposing report on Cav-1 expression in patients[32] performed non-statistical immunohistochemical analysis in 10 UC and 10 CD patients and observed a decreasing trend in Cav-1 expression. In contrast, the present analyses used two well-organized clinical datasets of patients with IBD (*n* = 190 and *n* = 194), resulting in statistically significant evaluations of Cav-1 expression (Fig. 1a). Moreover, clinical proteomic evaluation also supported the finding that Cav-1 is elevated in patients with IBD (Fig. 1b). In contrast with the adverse activity of Cav-1 in the present model, a previous investigation using a 2,4,6-trinitrobenzenesulfonic acid-induced colitis model suggested that Cav-1 played protective roles against intestinal inflammation with no molecular evidence of its effects[33]. 2,4,6-Trinitrobenzenesulfonic acid-induced colitis is T cell-dependently regulated, leading to Cav-1 suppression[33]. In contrast, we attempted to exclude the effects of acquired immunity and used a DSS-induced colitis model, which is mostly T cell-independent. Moreover, another study also demonstrated increased Cav-1

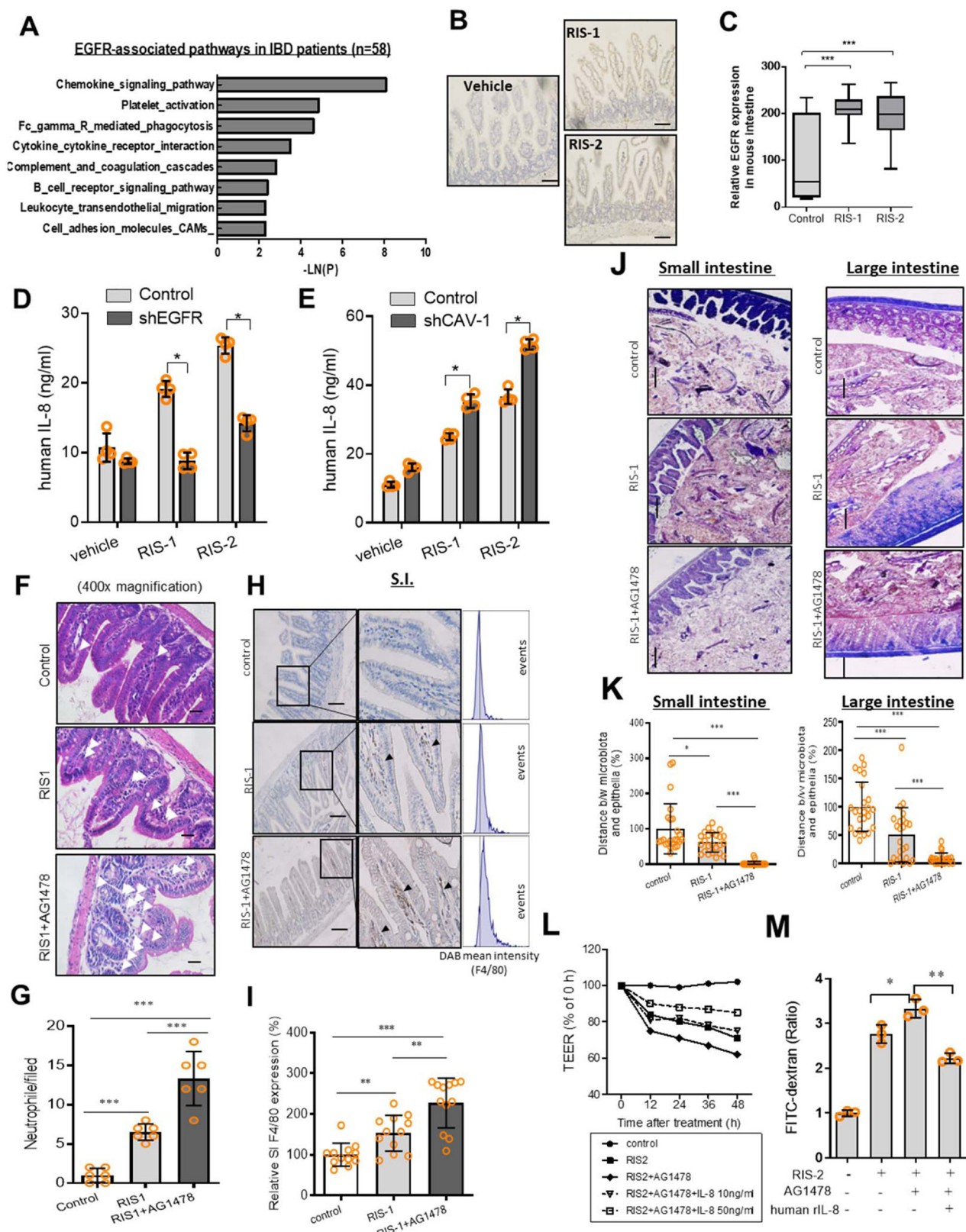

expression in a DSS-induced colitis model[34], which was consistent with the experimental results and clinical evaluation of datasets determined herein.

In the present study, the potential activity of Cav-1 was investigated based on analyses of the negative correlation between Cav-1 and EGFR in patients with IBD. Patient transcriptomic analysis suggested that EGFR was functionally involved in epithelial chemokine production, which was also mechanistically confirmed using the intestinal cell model. In addition to the well-defined chemokine activity in immune-related cell infiltration of mucosa, epithelial chemokines and their receptor-linked signals play pivotal roles in maintaining gut barrier integrity via signaling

**Fig. 4 Roles of EGFR-induced epithelial chemokines in the ribosome-inactivated gut barrier. a** KEGG-based functional annotations of EGFR-related pathways in patients with IBD (GEO ID: gse10616, $n = 58$). EGFR-correlated genes were functionally clustered into groups with each specific biological pathway, which can be collectively defined as EGFR-associated pathways. **b, c** EGFR expression was assessed in the ilea of 8-week-old male C57BL/6 mice treated with vehicle, 25 mg/kg RIS-1, or 25 mg/kg RIS-2 for 24 h ($n = 5$–7, each group). (**b**) Microscopic analysis was conducted at 200× magnification; scale bar(s): 100 μm. **c** Quantitative analysis is shown as box-and-whisker plots (min to max). **d, e** The IL-8 secretion. **d** Negative control or shEGFR-expressing HCT-8 cells were treated with vehicle, 50 ng/mL RIS-1, or 500 ng/mL RIS-2 for 12 h. **e** HCT-8 cells transfected with the negative control vector or shCav-1 were treated with vehicle, 50 ng/mL RIS-1, or 500 ng/mL RIS-2 for 12 h. Data represent the means ± SD ($n = 3$). **f–k** Eight-week-old male C57BL/6 mice received AG1478 (1 mg/mouse) intraperitoneally twice during 2 days before treatment with 25 mg/kg RIS-1 ($n = 5$–7 animals per group). Animals were sacrificed after 12 h of RIS-1 treatment. **f, g** Detection (**f**) and quantitation (**g**) of neutrophils in the small intestine based on segmented morphology in high-power fields (400×) using measurements from different randomly selected portal areas. **h,i** F4/80 staining in the small intestine (original magnification 200 ×; scale bar(s): 100 μm). (**i**) Quantitative comparisons are shown in the graph. **j, k** Gram staining to visualize the accessibility of the epithelial barrier by total bacteria (magnification: 100× (small intestine), 200 × (large intestine); scale bar(s): 100 μm). **k** Thickness of mucosal layers in the small and large intestines. **l, m** Cells were treated with combinations of RIS-2 (1000 ng/mL), AG1478 (5 μM), or human recombinant IL-8 (10 or 50 ng/mL). Epithelial permeability of differentiated HCT-8 cells exposed to each chemical combination was determined by measuring TEER (**l**) and the paracellular efflux of FITC-dextran at 48 h after treatment (**m**). (**a–m**) Asterisks (*) indicate significant differences between groups (*$p < 0.05$, **$p < 0.01$, and ***$p < 0.001$ using two-tailed unpaired Student's $t$ test).

cascades by boosting survival responses and wound repair in the insulted gut layer. In the present study, activation of barrier-protective EGFR contributed to epithelial chemokine production, which was positively involved in maintaining gut epithelial integrity rather than inflammatory cell recruitment. Even the suppression of EGFR signaling increased macrophage infiltration in the ribosome-inactivated gut layer. Disruption of the gut barrier by EGFR blockade allowed more microbial translocation and exposure to the underlying mucosal immune cells, leading to pro-inflammatory stimulation and subsequent recruitment of circulating macrophages to the inflamed gut tissues. This disrupted gut defense and bacterial exposure would lead to exaggerated inflammatory responses in patients with IBD. However, because recruited macrophages can damage inflamed gut tissues via detrimental cytokines, Cav-1-mediated downregulation of EGFR-linked cytokine production could be beneficial. In particular, myeloid EGFR is the pivotal trigger of harmful pro-inflammatory mediators in UC and colitis-associated cancers[35,36]. In contrast, it is well known that epithelial EGFR mediates the wound healing process and protects against ulcerative injuries in IBD and colitis-associated tumorigenesis[37,38]. Our results suggested that RIS-induced epithelial caveolar activities interfered with EGFR-mediated wound healing in the gut epithelia of patients with ulcerative injuries. Moreover, gut epithelial chemokines are important beneficial signals for immune cell recruitment to defend against infection in gut ulcers[39,40], which was consistently confirmed in the present study (Fig. 4e–i). EGFR-mediated chemokine production was shown to play pivotal roles in maintaining the chemical and physical barriers of the intestine against gut microbial invasion and subsequent inflammatory injuries[40]. Therefore, prolonged Cav-1 induction may retard epithelial restitution, resulting in further luminal infection in the gut epithelium of patients with IBD or colitis-associated malignancy. In addition to defending against infection, the gut mononuclear phagocytic system can play protective roles, such as wound healing and tissue restitution in response to intestinal injury, while suppressing inflammatory responses of early recruited neutrophils[41–43].

Cav-1, a key structural component of caveolae, associates with cavin-1 (also known as polymerase I and transcript release factor) as a soluble cytosolic protein, which protects Cav-1 from lysosomal degradation. This association between Cav-1 and cavin-1 in the plasma membrane is required for caveolar formation and sequestration of mobile caveolin into immobile caveolae[44,45]. Moreover, post-translationally modified cavin-1 can be involved in the caveolae-independent promotion of rRNA transcription, which is crucial to various physiological responses of cell growth

and metabolism[46]. In terms of caveolar actions, RTK-like orphan receptor 1 can serve as the pivotal scaffold protein for Cav-1, cavin-1, and cavin-3 in caveolae for endocytosis[47,48]. In turn, the RTK-like orphan receptor 1-linked endocytosis complex can activate downstream signaling, such as serine/threonine protein kinase B, of other RTKs, including EGFR in cancer cells. Therefore, disruption of this complex would elevate EGFR phosphorylation or expression as compensatory feedback. In contrast, excessive activation of the RTK-like orphan receptor 1-caveolar complex may retard activation signals from EGFR or reduce EGFR levels through negative feedback. Any of these possibilities could account for Cav-1-mediated negative regulation of EGFR in the present experimental models and clinical transcriptome datasets. In contrast with Cav-1, cavin-1 was not correlated with EGFR expression although cavin-1 was elevated in patients with IBD (unpublished data). Moreover, Cav-1 was shown to facilitate EGFR protein degradation via proteasomal activation. A 20-amino acid membrane-proximal region of the cytosolic $NH_2$-terminal domain of Cav-1, also known as the caveolin-scaffolding domain, mediates its interaction with EGFR[49]. This scaffolding domain stabilizes the inactive form of EGFR, which also contains a caveolin-binding motif in its cytoplasmic kinase domain. Because the caveolin-binding motif is conserved in most RTKs[49], Cav-1 can regulate diverse biological events in association with RTKs, as suggested with the ROR1-caveolar complex. However, these specific events are critically dependent on the types of stimuli and exposure patterns in different regimes of pathogenesis. After binding with Cav-1, the enzyme activities of EGFR can be altered via conformational changes in vitro;[49] however, in this study, promoted protein degradation was observed after binding in ribosome-inactivated cells. Taken together, the results presented herein indicate that negative EGFR regulation by Cav-1 can be explained by either protein degradation or enzymatic inhibition after binding between the caveolin-scaffolding domain and caveolin-binding motif. However, intact EGFR has a greater chance of transmitting stress signals into insulted enterocytes after dissociating from Cav-1, ultimately leading to chemokine production. In terms of gene regulation, the sub-cellular localization of EGFR can influence its signaling and cellular localization following activation. Specifically, activated EGFR localized in the plasma membrane tends to transmit both MAPK- and serine/threonine protein kinase B-associated signals, while endosomal EGFR activation results in ligand-dependent activation of ERK and p38 MAPK pathways[50,51]. Moreover, nuclear EGFR can act as a transcriptional regulator to induce several target genes[52,53], despite not directly binding to their promoters. In our stress model, ribosomal inactivation-induced Egr-1 was shown to act as

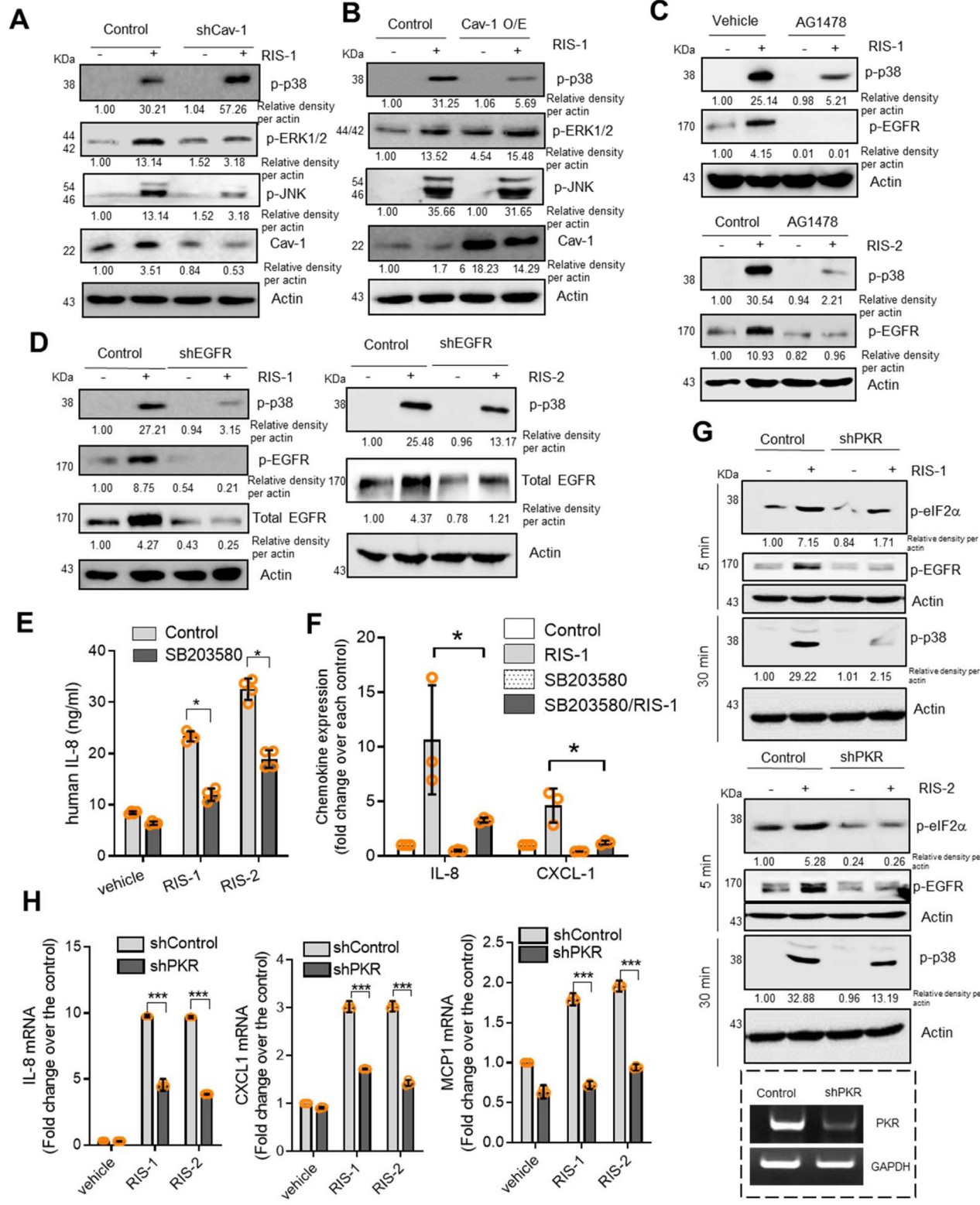

the crucial transcriptional factor in nuclear or cytoplasmic EGFR-regulated chemokine induction. However, further studies are warranted to investigate the mechanism of EGFR behavior involved in spatially regulating target genes in stressed cells.

Although EGFR was beneficial to the barrier defense against infection, repeated activation of growth-promoting signals such as EGFR-linked pathways after toxic or inflammatory insults would ultimately contribute to cancer cell survival and tumorigenesis progression[54,55]. Chronic mucosal insults persistently trigger EGFR-mediated oncogenic processes in intestinal epithelial cells, thus facilitating survival and metastasis of the transformed cells by providing a growth advantage[56]. It is estimated that EGFR is overexpressed in 60–80% of colorectal cancers, and its high expression is associated with a poor prognosis in patients

**Fig. 5 Effects of Cav-1 on downstream signaling mediators of EGFR in ribosome-inactivated intestinal cells. a** HCT-8 cells transfected with control or shCav-1 vector were treated with vehicle or 50 ng/mL RIS-1 for 30 min. Cell lysates were subjected to western blot analysis. **b** HCT-8 cells transfected with control or Cav-1 overexpression vector were treated with vehicle or 50 ng/mL RIS-1 for 30 min. Cell lysates were subjected to western blot analysis. **c** HCT-8 cells were pre-exposed to control or 10 μM AG1478 for 2 h and treated with vehicle, 500 ng/mL RIS-2, or 50 ng/mL RIS-1 for 30 min. Cell lysates were subjected to western blot analysis. (**d**) HCT-8 cells transfected with control or shEGFR vector were treated with vehicle, 500 ng/mL RIS-2, or 50 ng/mL RIS-1 for 30 min. Cell lysates were subjected to western blot analysis. (**e**) HCT-8 cells were pre-exposed to vehicle or 10 μM SB203580 for 2 h and treated with vehicle, 50 ng/mL RIS-1, or 500 ng/mL RIS-2 for 12 h. The IL-8 concentration secreted into culture media was measured by ELISA. **f** HCT-8 cells were pre-exposed to vehicle or 10 μM SB203580 for 2 h and treated with vehicle, 50 ng/mL RIS-1, or 500 ng/mL RIS-2 for 1 h. IL-8 and CXCL-1 mRNA was measured using real-time RT-PCR. **g** HCT-8 cells transfected with control or shPKR vector were treated with vehicle, 500 ng/mL RIS-2, or 50 ng/mL RIS-1 for 5 or 30 min. Cell lysates were subjected to western blot analysis. **h** HCT-8 cells transfected with control or shPKR vector were treated with vehicle, 500 ng/mL RIS-2, or 50 ng/mL RIS-1 for 1 h. IL-8, CXCL-1, and MCP1 mRNA was measured using real-time RT-PCR. **e–h** Asterisks (*) indicate significant differences between groups (*$p < 0.05$, **$p < 0.01$, and ***$p < 0.001$ using two-tailed unpaired Student's $t$ test).

with colorectal cancer[57]. In particular, exposure to dietary components such as high fat diet has been associated with the modulation of experimental intestinal cancers via an EGFR-mediated mechanism[58,59]. Therefore, optimal phase-dependent regulation of EGFR-linked signaling pathways could be crucial in preventing oncogenic processes in patients with chronic ulcerative gut injuries and inflammation. Therefore, both beneficial and detrimental activities of EGFR must be carefully assessed in chronically insulted enterocytes and transformed cells, although EGFR can also be involved in the maintenance of gut barrier integrity via protective factors against gut microbial invasion and subsequent inflammation. IBD is a complex disease that involves various etiological factors of genetic and environmental origin. In response to external or intracellular stressors, the ribosomal sensor was assessed as a potential trigger of caveolae-associated IBD pathogenesis in the present study. Cav-1 enhancement by ribosomal inactivation was negatively associated with EGFR-linked epithelial barrier integrity. The molecular evidence of crosstalk between ribosomes and caveolae provides comprehensive insight into the organelle-mediated regulatory network of pathological events in malignant intestinal diseases, including IBD and colorectal cancer.

Clinical dataset-based analysis in patients with IBD indicated a remarkably elevated expression of PKR and stress responsive genes in a PKR-dependent manner. PKR-linked biological responses were simulated in experimental gut models of ribosome-inactivating stress. However, mammalian PKR is activated mainly by double-stranded RNA (dsRNA) produced during viral infection[60–62]. In addition to dsRNA, single-stranded RNA also binds to PKR and causes a conformational change, leading to its dimerization via its C-terminal kinase domain and subsequent autophosphorylation during viral infection[61,63]. Activated PKR inhibits viral and host protein synthesis through eIF2α phosphorylation, which facilitates the stress responses and antiviral defense. Moreover, PKR can be activated by other diverse stresses, such as oxidative and endoplasmic reticulum stress, metabolic stress, or by other chronic inflammatory stresses in a dsRNA-independent manner[12,64–66]. Therefore, although we focused on a ribosomal stress-based assessment of PKR-linked events, additional evaluations are warranted in pathologic states, including viral infection, tumorigenesis, and diet-associated metabolic stress. Such extensive assessment would explain the complex PKR-linked translational network in the patients experiencing gut distress throughout their lives.

## Methods
**Cell culture and chemical treatment**. HCT-8, SW480, and HT29 cells were purchased from the American Type Culture Collection (ATCC, Manassas, VA, USA). Cells were maintained in RPMI 1640 medium (Welgene, Daegu, South Korea) supplemented with 10% [v/v] heat-inactivated fetal bovine serum (Welgene), 25 mM HEPES, 50 U/mL penicillin, and 50 mg/mL streptomycin (Welgene)

in a 5% $CO_2$ humidified incubator at 37 °C. Cells were enumerated by trypan blue (Sigma-Aldrich, St. Louis, MO, USA) dye exclusion using a hemocytometer. All chemicals were purchased from Sigma-Aldrich. HCT-8 cells were pre-exposed to each inhibitor (2 μM U0126, 10 μM SB203580, 10 μg/mL MDC, 50 μg/mL Nystatin or 10 μM SP600125) for 2 h, and then treated with 50 ng/mL RIS-1 or 500 ng/mL RIS-2 for indicated times.

**Mouse experiments**. C57BL/6 mice (6 weeks old, 16–18 g on average) were purchased from Jackson Laboratories (Bar Harbor, ME, USA). Mice were acclimated for 14 days prior to experiments and maintained at 25 °C in 45–55% relative humidity under 12 h light/dark cycles. Mice were housed three per cage and provided sufficient food and water in environmentally protected cages comprising a transparent polypropylene body and a stainless-steel wire top cover. Animal care and experimental procedures were conducted in accordance with our Institutional Animal Care and Use Committee's guidelines. This animal study was approved by the Pusan National University Institutional Animal Care and Use Committee (PNU-IACUC) (PNU-2015-0786). We used chemical ribosomal inactivators such as anisomycin and deoxynivalenol, which specifically inhibit or cleave 28S ribosomal RNA at the peptidyl transferase center, leading to ribosomal stalling and global translational inhibition[16–19]. For in vivo ribosomal inactivation, 8-week-old male mice were exposed to 25 mg/kg anisomycin (ANS, RIS-1) or deoxynivalenol (DON, RIS-2) via oral gavage with a 4 cm-long curved ball-tip needle. At 12 or 24 h after chemical treatment, the intestines were isolated and fixed in neutral-buffered 4% paraformaldehyde solution. To determine the effects of EGFR inhibition, mice were pretreated with 1 mg AG1478 (Selleckchem, Houston, TX, USA) or an equal volume of vehicle control per mouse twice during 2 days (the second administration was at 2 h before RIS-1 treatment). The mice were then exposed to 25 mg/kg RIS-1 for 12 h and subsequently sacrificed for histological observation. To induce IBD-like symptoms in the colon, 8-week-old female mice were administered 3% dextran sodium sulfate (DSS, $M_w$ 36,000–50,000 Da; MP Biomedical, Solon, OH, USA) ad libitum in drinking water for 5 days. Mice were then sacrificed under deep ether anesthesia after another 5 days. Post-mortem, the colons were excised, washed gently with phosphate-buffered saline (PBS) to remove fecal debris, rolled into a Swiss-roll formation, and fixed in neutral-buffered 4% paraformaldehyde solution.

**Analysis of colonic mRNA and proteins of patients with IBD**. Human colonic tissue datasets were obtained from gene expression arrays of patients with IBD (gse117993 [Denson's, $n = 190$] and gse75214 [Vemeire's, $n = 194$]). For clinical dataset, the three major clinical subsets of IBD included only cCD, iCD, and UC. These experiments tested differential colonic gene expression in these three types of IBD relative to healthy control samples. Protein levels from the colonic biopsies were compared using the proteomic dataset of patients with IBD (PXD005086, $n = 99$).

**Caenorhabditis elegans model**. Caenorhabditis elegans Bristol N2 (Brenner 1974), caveolin-1 mutant (Cav-1 MT, genotype cav-1[ok2089] IV), and EGFR mutant (let-23 MT, genotype let-23[n1045] II) strains were obtained from Caenorhabditis Genetics Center, University of Minnesota, Minneapolis, MN, USA and maintained at 20–25 °C on nematode growth medium (NGM) agar plates (50 mM NaCl, 1.7% agar, 0.25% peptone, 1 mM $CaCl_2$, 5 μg/mL cholesterol, 1 mM $MgSO_4$, and 25 mM $KPO_4$ in $dH_2O$) spread with Escherichia coli OP50 ($OD_{600} = 0.6$–0.8) as the food source. Caenorhabditis elegans were synchronized with a mixture of 500 μL 5 N NaOH, 1 mL 5% NaOCl (Yohanclorox, Seoul, South Korea), and 3.5 mL autoclaved $dH_2O$. Synchronized eggs were then seeded on NGM plates for growth, and L4-stage worms were seeded onto new NGM plates containing 50 μM 5-fluorodeoxyuridine and E. coli OP50 ($OD_{600} = 0.6$–0.8). For experimental treatments, chemicals from stock solutions were added and spread onto culture plates. For DAPI staining, worms were washed in M9 buffer on a slide, wiped with Whatman paper to remove extra liquid, and washed three times with 95% ethanol;

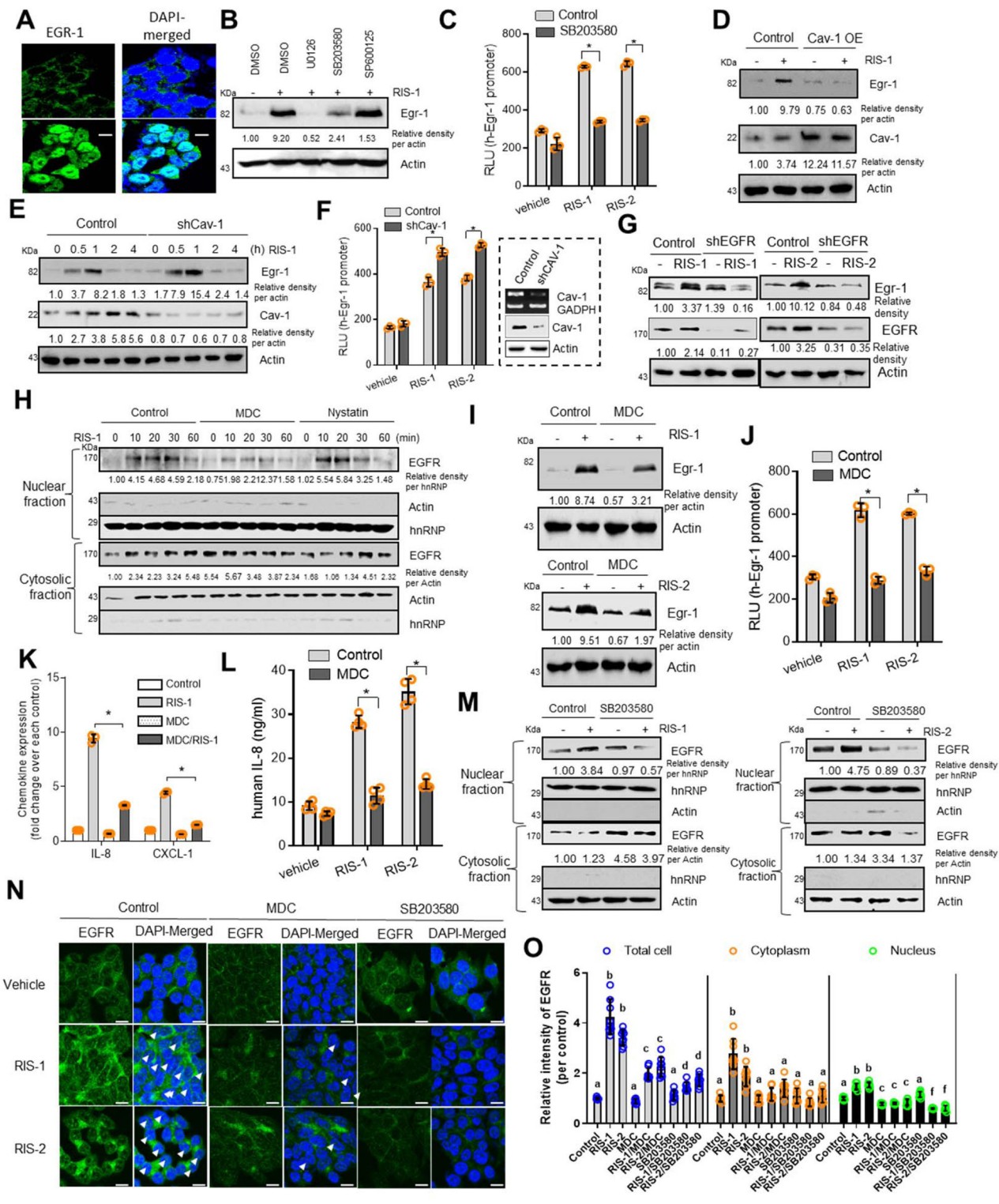

fixed worms were then treated with mounting solution containing DAPI (IHC World, Ellicott City, MD, USA). To prepare the bacterial colonization assays, worms were age-synchronized by bleaching, and embryos were incubated at 24 °C on NGM plates containing *E. coli* OP50 until the L4 stage. Following exposure to DON for 24 h, 10 worms were selected and placed in M9 buffer containing 25 mM levamisole for paralysis and inhibition of pharyngeal pumping. The animals were then treated with 100 μg/mL gentamicin and incubated for 45 min to remove surface bacteria. Animals were washed three times in LM buffer to remove bacteria and antibiotics, immersed in LM buffer containing 1% Triton X-100, and then mechanically disrupted using a motor and pestle for 3 min. Lysates of worms were subsequently diluted in M9 buffer, spread on LB agar containing 50 μg/mL

streptomycin, and incubated overnight at 37 °C. The gut bacterial colonies were quantified per nematode.

**Intestinal barrier function assay (Smurf assay)**. After 24 h of DON treatment, worms from the NGM plates were cultured in liquid media of UV-killed *E. coli* OP50 mixed with 5.0% wt/vol blue food dye (FD&C Blue #1, B0790, TCI Chemicals, Portland, OR, USA) in NGM liquid solution for 3 h. Animals were then washed in M9 buffer until the blue color of the dye was no longer visible and subsequently anesthetized in M9 buffer containing 25 mM levamisol. The worms were then evaluated for the presence of blue food dye in the body cavity using an

**Fig. 6 Regulation of EGR-1 and EGFR translocation by Cav-1. a** HCT-8 cells were treated with vehicle or 500 ng/mL RIS-2 for 2 h. Egr-1 was visualized (1800 × magnification; scale bar(s): 20 μm). **b–m** Chemical-treated HCT-8 cells (all treatment dosages depicted in *Methods*) were assessed for western blot analysis (**b, d, e, g–i, m**), luciferase assay (**c, f, j**), real-time RT-PCR (**k**), and IL-8 ELISA (**l**). **b** HCT-8 cells pre-exposed to each inhibitor were treated with vehicle or RIS-1 for 1 h. **c** Cells pre-exposed to SB203580 were treated with vehicle or RIS-2 for 9 h. **d** Transfected cells were treated with vehicle or RIS-1 for 1 h. **e** Transfected cells were treated with vehicle or RIS-1 for the indicated time. **f** Transfected cells were treated with vehicle, or each RIS for 9 h. **g** Transfected cells were treated with vehicle, or each RIS for 1 h. **h** Cells pre-exposed to each inhibitor were treated with vehicle or RIS-1 for the indicated time. **i** Cells pre-exposed to each inhibitor were treated with vehicle, RIS-1, or RIS-2 for 1 h. **j** Transfected cells pre-exposed to each inhibitor were treated with vehicle, RIS-1, or RIS-2 for 9 h. **k** Cells pre-exposed to each inhibitor were treated with vehicle or RIS-1 for 1 h. **l** Cells pre-exposed to each inhibitor were treated with vehicle, RIS-1, or RIS-2 for 12 h. **c, f, j–l** Asterisks (*) indicate significant differences between groups (*$p < 0.05$, **$p < 0.01$, and ***$p < 0.001$ using two-tailed unpaired Student's $t$ test). **m** Cells pre-exposed to each inhibitor were treated with vehicle, RIS-1, or RIS-2 for 20 min. **n, o** HCT-8 cells pre-exposed to each inhibitor were treated with vehicle, RIS-1, or RIS-2 for 20 min to detect nuclear EGFR (1800 ×; scale bar(s): 20 μm). **o** Different letters over each bar in the quantitative analysis represent significant differences between groups in each compartment (the whole cell, cytoplasm, or nucleus) ($p < 0.05$ using one-way ANOVA with the Newman–Keuls post hoc test).

Eclipse Ts2R microscope (Nikon, Tokyo, Japan) under 20 × magnification. Data were analyzed using GraphPad Prism.

**Measurement of transepithelial electrical resistance (TEER)**. HCT-8 cells were seeded at a density of $4 \times 10^5$ cells/well in 24-well transwell filters with 0.4 μm pores (Becton-Dickinson Labware, Franklin Lakes, NJ, USA). After reaching confluence within 2 days, cells were treated with complete RPMI1640 and Dulbecco's modified Eagle's medium/F-12 media containing 100 nM dexamethasone, which was changed every other day until complete differentiation occurred. At the end of the differentiation process (on day 10 after dexamethasone addition), cells were co-treated with 1000 ng/mL RIS-2 (DON), EGFR inhibitor, or human recombinant IL-8. TEER was then measured every 12 h with an EVOM2 epithelial voltohmmeter (World Precision Instruments, Sarasota, FL, USA). Experimental TEER values were expressed as $\Omega \times cm^2$. All measurements were made using three replicates from each experiment.

**Paracellular tracer flux assay**. Differentiated HCT-8 cells in 0.4 μm pore inserts were treated with various combinations of chemicals (1000 ng/mL RIS-2, 5 μM AG1478, and 10 or 50 ng/mL human recombinant IL-8). After 48 h of treatment, 4 kDa fluorescein isothiocyanate-dextran (FITC-dextran, Sigma-Aldrich, St. Louis, MO, USA) was added to the apical compartment of enterocytes at a final concentration of 2.2 mg/mL. After 1 h of incubation, the fluorescence in the basolateral compartment was measured using a Victor3 fluorometer (Perkin Elmer, Waltham, MA, USA) at excitation and emission wavelengths of 490 and 535 nm, respectively. The data herein are representative of three independent experiments.

**Plasmid construction**. The cDNA containing the entire coding region of N-terminal flag-tagged Cav-1 was generated by reverse transcription and polymerase chain reaction (RT-PCR) using RNA from HCT-8 cells with the following primers: 5′-CAC CAT GGA CTA CAA GGA CGA CGA TGA CAA GAT GGC AGA CGA GCT GAG CGA-3′ (forward) and 5′-TTA TAT TTC TTT CTG CAA GT-3′ (reverse). The resulting 475-bp construct was then cloned using a TopCloner TA Kit (Enzynomics, Daejeon, South Korea), excised at the EcoRI sites, and then transferred into the expression plasmid pcDNA3.1 (−) in the sense orientation (Thermo Fisher Scientific, Waltham, MA, USA) using T4 DNA ligase (NEB, Ipswich, MA, USA). This vector was denoted as Cav-1 O/E. C-terminal flag-tagged ubiquitin (ubiquitin O/E-flag) was kindly provided by Koji Nakagawa (Hokkaido University, Sapporo, Japan), and plasmids containing the human Egr-1 promoter (−1.26 kb)-linked reporter gene was kindly gifted from Thomas Eling (National Institute of Environmental Health Sciences, Durham, NC, USA). CMV-driven small interfering RNA was generated by inserting short hairpin RNA (shRNA) templates into a pSilencer 4.1-CMVneo vector (Ambion/Thermo Fisher Scientific, Seoul, Korea). The negative control vector, Cav-1 shRNA, and EGFR shRNA were denoted as control, shCav-1, and shEGFR, respectively. The template sequence of negative control siRNA lacking homology to the mouse, human, and rat genome databases and pSilencer 4.1-CMV neo containing the negative control siRNA template were obtained from Ambion/Thermo Fisher Scientific. Cav-1, PKR, and EGFR shRNA targeted the sequences 5′-GCC CAA CAA CAA GGC CAT G-3′, 5′-GCG AGA AAC TAG ACA AAG T-3′, and 5′-CTC TGG AGG AAA AGA AAG T-3′, respectively. The entire coding region of ATF3, including the TATAA region, was generated by RT-PCR using RNA from HCT-8 cells with the following primers: forward 5′-CGT GAG TCC TCG GTG CTC-3′, reverse 5′-GAC AGC TCT CCA ATG GCT TC-3′. The resulting 721-bp construct was cloned using the TopCloner TA Kit (Enzynomics), excised at the HindIII/NotI sites, transferred in sense and antisense orientations into the expression plasmid pcDNA3.1Zeo + /− (Invitrogen, Carlsbad, CA, USA) using T4 DNA ligase (NEB, Ipswich, MA, USA), and then confirmed by DNA sequencing for antisense ATF3.

**Transient transfection**. Cells were transfected with a mixture of plasmids using jetPRIME (Polyplus Transfection, New York, NY, USA) or OmicsFect (Omicsbio, Taipei, Taiwan) according to the manufacturer's protocols to achieve transient expression of the control, Cav-1 O/E, ubiquitin O/E-Flag, shCav-1, and shEGFR. Transfection efficiency was confirmed by expression of a pMX-GFP vector. At 5 h after transfection, the media were changed and the cells were incubated for another 48 h. For OmicsFect transfection, the cell culture media were changed after 24 h.

**Conventional and real-time RT-PCR**. Total RNA was extracted with RiboEx (GeneAll Biotech, Seoul, South Korea) according to the manufacturer's instructions, after which RNA (500 ng) from each sample was transcribed to cDNA using Prime RT premix (Genetbio, Nonsan, South Korea). The cDNA was amplified using Takara HS ExTaq DNA polymerase (Takara Bio, Shiga, Japan) in a MyCycler Thermal Cycler (Bio-Rad, Hercules, CA, USA) using the following parameters: denaturation at 94 °C for 2 min followed by 25 cycles of denaturation at 98 °C for 10 s, annealing at 59 °C for 30 s, and elongation at 72 °C for 45 s. An aliquot of each PCR product was subjected to 1.2% (w/v) agarose gel electrophoresis and visualized by staining with ethidium bromide. The primers for PCR are shown in the supplementary information (Supplementary Table 1). For real-time PCR, FAM was used as a fluorescent reporter dye and conjugated to the 59 ends of the probes used to detect the amplified cDNA. Real-time PCR was performed with Rotor-Gene Q (Qiagen, Hilden, Germany) using the following parameters: denaturation at 94 °C for 2 min followed by 40 cycles of denaturation at 98 °C for 10 s, annealing at 59 °C for 30 s, and elongation at 72 °C for 45 s. Each sample was tested in triplicate to ensure statistical significance. Relative quantification of gene expression was performed using the comparative Ct method, where the Ct value was defined as the point at which a statistically significant increase in fluorescence was observed. The number of PCR cycles (Ct) required for the FAM intensities to exceed a threshold value just above the background level was calculated for the test and reference reactions. In all experiments, GAPDH was used as the internal control. The results were analyzed relative to vehicle-treated samples.

**Western blot analysis**. Protein expression levels were compared by western blot analysis. Briefly, cells were washed with ice-cold phosphate buffer, lysed by boiling in lysis buffer (1% [w/v] SDS, 1.0 mM sodium ortho-vanadate, and 10 mM Tris [pH 7.4]), and sonicated for 5 s. Proteins in the lysates were quantified using a WelProt Protein Assay Kit (Welgene, Daegu, South Korea). In total, 20 μg protein was separated by electrophoresis using a Bio-Rad mini gel apparatus (Bio-Rad, Hercules, CA, USA). The proteins were then transferred to a polyvinylidene difluoride membrane (Pall Corporation, New York City, NY, USA) and blocked for 1 h with 5% skimmed milk in TBS plus 0.1% Tween 20 (TBST) before probing with primary antibodies (diluted at 1:1000) for 2 h at room temperature or overnight at 4 °C. The primary antibodies used were rabbit polyclonal anti-β-actin, anti-Cav-1, anti-Egr-1, anti-EGFR, anti-phospho-p38, mouse monoclonal anti-hnRNP and anti-phospho ERK1/2 (Santa Cruz Biotechnology, Santa Cruz, CA, USA), and rabbit polyclonal anti-phospho-EGFR (Epitomics, Burlingame, CA, USA). Anti-flag antibody was purchased from Sigma-Aldrich. After washing three times with TBST, the blots were incubated with horseradish-conjugated secondary antibody for 1 h, then washed three more times with TBST. Finally, antibody binding was detected with Pico enhanced peroxidase detection (ELPIS Biotech, Daejeon, South Korea).

**Isolation of cellular fractions**. For nuclear protein isolation, cells were scraped into ice-cold PBS. After centrifugation, the collected cell pellet was then resuspended in lysis buffer containing 10 mM HEPES, 10 mM KCl, 1.5 mM MgCl₂, 0.5 mM dithiothreitol, 0.5 mM phenylmethylsulfonyl fluoride, 0.1% Nonidet P-40, and a protease inhibitor mixture (Sigma-Aldrich); then, the cells were incubated for 10 m on ice and centrifuged. The supernatant (the cytosolic fraction) was collected, and the pellet was resuspended in buffer containing 20 mM HEPES, 1.5 mM

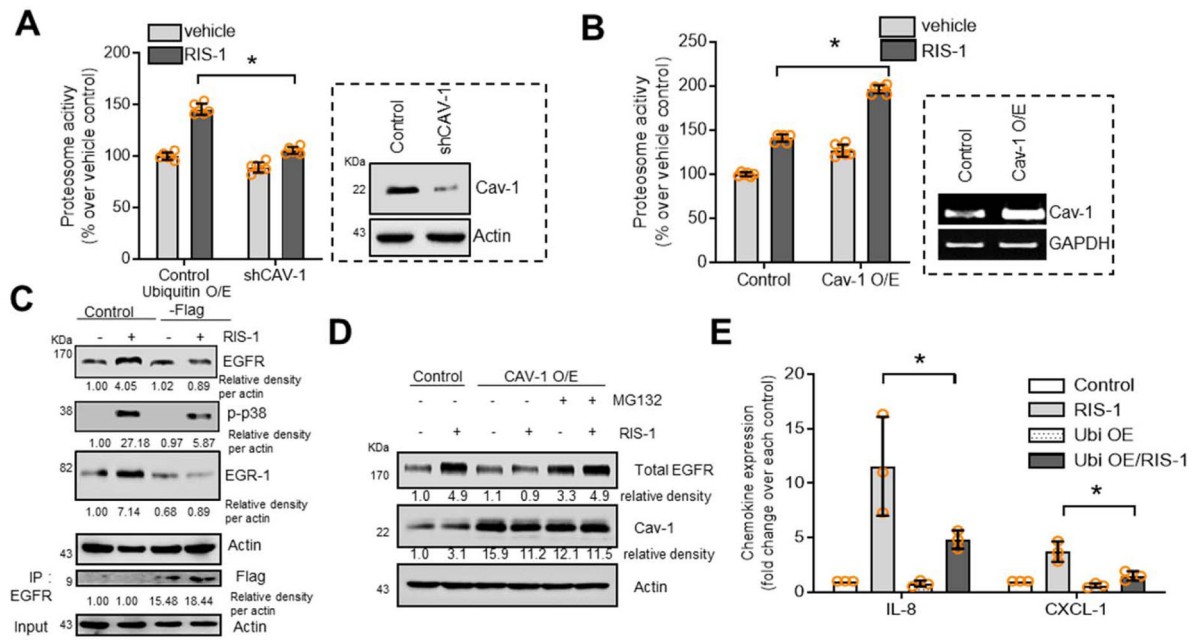

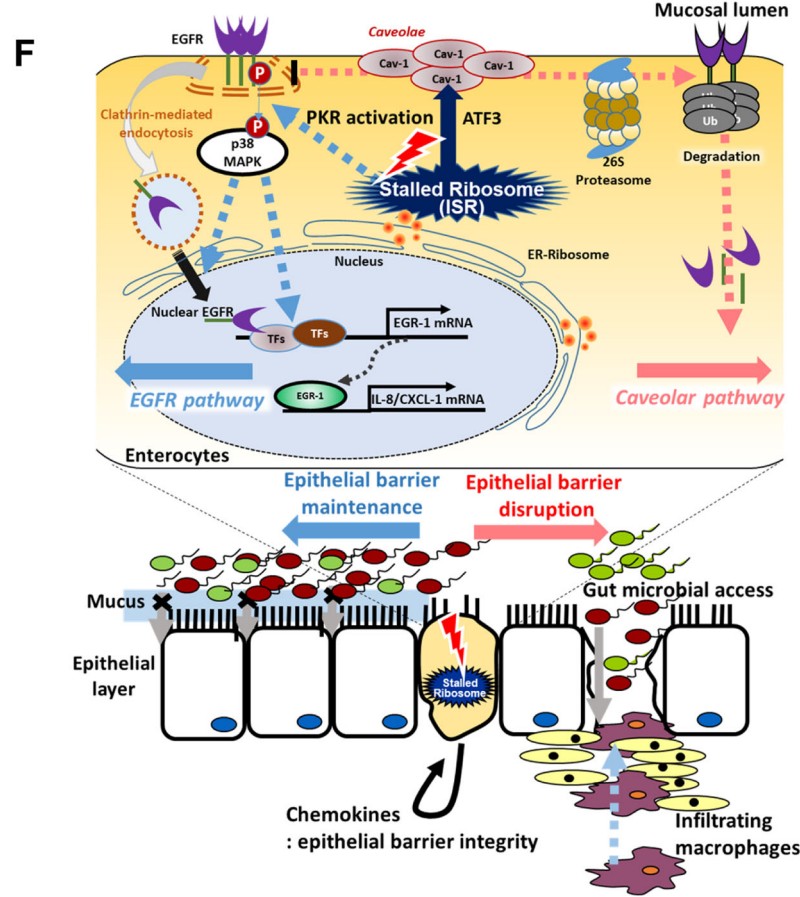

MgCl$_2$, 420 mM NaCl, 0.2 mM EDTA, 0.5 mM dithiothreitol, 0.2 mM phenylmethylsulfonyl fluoride, 25% glycerol, and a protease inhibitor mixture (Sigma-Aldrich). After incubating for 10 m on ice, the samples were centrifuged, and the supernatants (nuclear proteins) were collected and stored at −80 °C until analysis.

**Immunoprecipitation assay.** Cellular lysate was prepared in immunoprecipitation lysis buffer (50 mM Tris (pH 7.2), 150 mM NaCl, 1 mM EDTA, 400 mM Na$_3$VO$_4$,

and 2.5 mM phenylmethylsulfonyl fluoride) containing 0.1% NP-40, and then incubated on ice for 40 min. The lysates were subsequently centrifuged at 13,800 × g for 10 min, after which the supernatant was transferred into a new tube, treated with 2 µg antibody, and rotated overnight at 4 °C. Protein G-Sepharose in the antibody-cell lysate mixture was incubated by rotating at 4 °C for 3 h, after which immunoprecipitates were collected by centrifugation and subjected to SDS-PAGE.

**Fig. 7 Regulation of EGFR stability by Cav-1 in ribosome-inactivated intestinal cells. a** Control- or shCav-1-transfected HCT-8 cells were treated with vehicle or 50 ng/mL RIS-1 for 20 min and subjected to proteasome activity assay. **b** Control vector- or Cav-1-overexpressing HCT-8 cells were treated with vehicle or 50 ng/mL RIS-1 for 20 min; then, cells were subjected to proteasome activity assay. **c** Control- or Flag-tagged ubiquitin-overexpressing HCT-8 cells were treated with vehicle or 50 ng/mL RIS-1 for 20 min (total EGFR and Flag), 30 min (p-p38), and 1 h (Egr-1). Cell lysates were subjected to western blot analysis. Total lysates were immunoprecipitated with anti-EGFR antibody and subjected to western blot analysis. (**d**) Control- or Cav-1-overexpressing HCT-8 cells were pre-exposed to vehicle or 20 μM MG132 for 6 h and treated with vehicle or 50 ng/mL RIS-1 for 20 min. Cell lysates were subjected to western blot analysis. **e** Control- or Flag-tagged ubiquitin-overexpressing HCT-8 cells were treated with vehicle or 50 ng/mL RIS-1 for 1 h. IL-8 and CXCL-1 mRNA was measured using real-time RT-PCR. **a**, **b**, **e** Asterisks (*) indicate significant differences between groups (*$p < 0.05$, **$p < 0.01$, and ***$p < 0.001$ using two-tailed unpaired Student's $t$ test). Results and in vitro evaluations are representative of three independent experiments. **f** A putative scheme for Cav-1-mediated barrier disruption in the ribosome-inactivated gut. In response to ribosomal inactivation, ATF3-mediated Cav-1 production in caveolae counteracts the PKR- and EGFR-activated signaling pathways in intestinal epithelial cells. Activated caveolar clustering downregulates clathrin-dependent EGFR translocation and barrier-protective chemokine production. Disruption of PKR-EGFR-chemokine-mediated maintenance of barrier integrity allows microbial access into inner tissues, leading to further pro-inflammatory stimulation, including macrophage infiltration.

---

**ELISA.** Cell debris was removed from the culture media by centrifugation; then, supernatant IL-8 concentrations were measured using an OptEIA Human IL-8 ELISA Kit (BD Biosciences, Franklin Lakes, NJ, USA) according to the manufacturer's instructions. Briefly, the ELISA plates were coated with capture antibody overnight at 4 °C, washed with PBS containing Tween-20, blocked overnight with PBS supplemented with 10% (v/v) fetal bovine serum at 4 °C, and then incubated with serial dilutions of the samples and IL-8 standards. Following treatment with detection antibody and tetramethylbenzidine substrate, the absorbance was measured at 405 nm using an ELISA reader. The assay detection limit was 3.1 pg/mL IL-8.

**Fluorogenic substrate proteasome activity assay.** HCT-8 cells were collected and resuspended in 200 μL of proteasome activity assay buffer (115 mM NaCl, 1 mM $KH_2PO_4$, 5 mM $CaCl_2$, 1.2 mM $MgSO_4$, 25 mM sodium HEPES buffer (pH 7.4)) containing 62.5 mM Suc-Leu-Leu-Val-Tyr-AMC (Suc-LLVY-AMC; Enzo Life Sciences, Farmingdale, NY, USA). The same quantity of cell lysate was incubated at 37 °C for 1 h and then measured at 360 nm/460 nm using a Victor3 Multilabel Plate Reader (Perkin Elmer, Waltham, MA, USA).

**Immunohistochemistry.** Formalin-fixed paraffin-embedded tissues were cut, deparaffinized, and rehydrated; then, the tissue sections were heated in 10 mM sodium acetate (pH 9.0) for 5 min at 121 °C for antigen retrieval. To remove endogenous peroxidase, tissues were soaked in 1% (v/v) $H_2O_2$-PBS solution for 15 min at room temperature in the dark. Next, samples were washed with 0.1% TBS-T, blocked with 3% (w/v) fetal bovine serum in PBS for 1 h, and then incubated with primary antibodies (1:200 dilution) overnight at 4 °C. After washing three times with 0.1% TBS-T, samples were incubated with horseradish peroxidase-conjugated secondary antibody for 2 h at room temperature and then washed with 0.1% TBS-T three times. The bound antibodies were identified using freshly prepared substrate buffer (0.05% [w/v] diaminobenzidine (DAB; Sigma-Aldrich) and 0.01% [v/v] $H_2O_2$ in PBS) for 5 min. After a final wash in distilled water, the sections were counterstained with 20% (v/v) hematoxylin (Santa Cruz Biotechnology) solution for 1 min and then dehydrated. Sections were subsequently examined at various magnifications using an Eclipse Ts2R (Nicon Instruments Inc., Melville, NY, USA). To analyze the proportion of positively stained cells in the tissue, at least four representative areas were measured by computer-assisted analysis using HistoQuest image analysis software version 4.0 (TissueGnostics, Vienna, Austria). The pixels in images viewed in this software were converted to grayscale and assigned an arbitrary number relating to staining intensity. The HistoQuest image analysis software differentiates hematoxylin-positive and DAB-positive cells, areas that are recorded as events (cells/nuclei), or areas of staining. The results are presented as histograms to which staining areas or intensity cutoffs can be applied to differentiate between cell populations and to determine the number of expressing cells and expression level of a given marker. Values were optimized by counting the numbers of DAB-positive events and hematoxylin-positive events.

**Confocal microscopy.** Cells were fixed with 4% paraformaldehyde (Biosesang, Sungnam, South Korea), permeabilized with 0.2% Triton X-100 in PBS for 10 min, and then blocked with 3% BSA in PBS for 2 h. The cells were then incubated at room temperature for 2 h with target antibody (1:200 dilution) in buffer (3% BSA in PBS), washed with PBS three times, and incubated with Alexa Fluor 488 goat anti-rabbit IgG (H + L) at room temperature for 2 h. Next, the cells were washed again in PBS and subsequently stained with 500 ng/mL DAPI (absorbance at 405 nm) in PBS for 5 min. Confocal images were obtained with an Olympus FV1000 confocal microscope (Olympus, Tokyo, Japan) using single-line excitation (488 nm for anti-rabbit antibodies, or 546 or 596 nm for anti-mouse antibodies) or multitrack sequential excitation (488 nm and 633 nm). Images were acquired and processed with the FV10-ASW software.

**Luciferase assay.** Cells were washed with cold PBS, lysed with passive lysis buffer (Promega, Madison, WI, USA), and then centrifuged at 12,000 × g for 15 min. The supernatant was subsequently collected, isolated, and stored at −80 °C until it was assessed for luciferase activity. A Model TD-20/20 dual-mode luminometer (Turner Designs, Sunnyvale, CA, USA) was used to measure the luciferase activity, and *firefly* luciferase activity was normalized against *Renilla* luciferase activity.

**Statistics and reproducibility.** Statistical analyses were performed using GraphPad Prism v. 5.01 (La Jolla, CA, USA). For comparative analysis of two groups of data, Student's $t$ test was performed. For comparative analysis of multiple groups, data were subjected to analysis of variance (ANOVA) with *Newman–Keuls* method as a post hoc ANOVA assessment. For two gene correlation coefficient (R) determination in IBD-based datasets, *Pearson*'s correlation analysis was performed. All in vitro evaluations are representative of two or three independent experiments. Details of the number of biological replicates and the assays are given in each figure legends.

**Reporting summary.** Further information on research design is available in the Nature Research Reporting Summary linked to this article.

## Data availability

All data needed to evaluate the conclusions in the paper are present in the paper. Additional data related to this paper are available from the corresponding author on reasonable request. The source data underlying plots shown in figures are provided in Supplementary Data 1. Full blots are shown in Supplementary Information.

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

## Acknowledgements

This research was supported by the Basic Science Research Program through the National Research Foundation of Korea (NRF) funded by the Ministry of Education (2018R1D1A3B05041889). We appreciate the technical assistance with worm and murine models provided by Mira Yu and Ray Navin.

## Author contributions

Project design and hypotheses were developed by Y.M. S.P. and J.K. conducted experiments and analyzed the data. Y.M. prepared the manuscript and supervised the overall project.

## Competing interests

The authors declare no competing interests.
