## [Peer Review File · Communications Biology]

Reviewers' comments:

Reviewer #1 (Remarks to the Author):

In this manuscript, park S et al showed that in IBD ribosome inactivation triggered caveolar clustering and activation which counteracted EGFR mediated signaling pathway and gut epithelial protection. This may provide a new insight into the pathogenesis of IBD. However, the results presented in this manuscript are not convincing to draw this conclusion, and there are multiple questions/concerns need to be further addressed.

Major concerns:

1. The authors assumed that there presented ribosome inactivation in IBD, and did all the experiments based on this hypothesis. However, this manuscript did not provide evidence/results to show the presence of ribosome inactivation in IBD patients and/or murine colitis.
2. The authors showed the gene expression level of Cav-1 and EGFR in IBD. How about their protein expression levels? This is important to support the authors' hypothesis.
3. To induce ribosome inactivation, the authors used RIS-1 in some experiments, and used RIS-2 in other experiments, and used both RIS-1 and RIS-2 in some experiments. What makes the authors to decide to use either RIS-1 or RIS-2? Was it due to inconsistent results caused by RIS-1 and RIS-2?
4. Previous publications showed that Cav-1 is not increased in the inflamed mucosa of patients with ulcerative colitis (Andoh, et al. *Inflammatory Bowel Disease*, 2001, 7(3):210-214), and Cav-1 was decreased in murine colitis and showed protective role for murine colitis (Weiss C, et al. *Plos One*, 2015). However, in the present study, the authors showed the increased Cav-1 expression in IBD patients. How will the authors explain the differences? It should be discussed in the discussion.
5. This manuscript is not well written. The purposes of some chemical treatment were not clearly mentioned. There were numerous English grammars found in this manuscript, and some part was hard to understand. Strongly suggest the authors to ask a Native English speaker to polish the English language of this manuscript.
6. The majority of in vitro assay was based on HCT-8 cell line. Did the authors try other human intestinal cell line and get similar findings?
7. In Fig 3, the authors showed the epithelial chemokine production was decreased by EGFR inhibition, but increased the neutrophil and macrophage recruitment. What's the reason to cause this? What's the role of EGFR-induced chemokine?

Minor concerns:

1. Does the expression of Cav-1 correlate with the severity of inflammation?
2. Did you perform the correlation analysis on the expression levels of Cav-1 and EGFR? Although Fig 1E kindly supported that Cav-1 expression is negatively correlated with EGFR expression in IBD.
3. In Figure legends it missed the repeating number of experiments performed or number used.

4. Figure legends were not well prepared.
5. Some figures miss scale bar. eg, Fig 3G
6. Fig 5G, the statistical significance among groups is not clear.

Reviewer #2 (Remarks to the Author):

Understanding the cellular and molecular principles by which tissues are organized, to ensure their integrity and repair, is essential to understand disease. In this report, Park and colleagues contribute insights onto how general cell stress mechanisms (the Integrated Stress Response) rewires the compartmentalization of the essential Egr signalling network, in a manner dependent on caveolae, to determine both epithelium activation and repair and the establishment of inflammation.

Conceptually, their study is interesting, in part because of the effort they make to overarch and integrate several mechanisms among each other. While it is not explicitly stressed in the report, they actually build their hypothesis from patient-derived gene expression datasets. I personally consider many ideas they propose of interest to a broad readership. There are however a number of points that the authors should address and discuss, mostly pertaining to their findings on Cav1 biology and ISR regulation.

MAJOR COMMENTS:

- A first major concern is the assumption that pharmacological stalling of polysomes should faithfully reproduce the events and dynamics downstream the activation of any branch of the ISR, which involves dysassembly of polysomes, potentially in a rather compartmentalized manner. Do they observe downstream hallmarks of ISR activation, such as upregulation of ATF4 or CHOP? I can understand the aim of overarching all those different "modalities" of ISR activation, but the authors should at the very least discuss the comparison of both conditions, and provide more extensive literature on the use of these ribosomal poisons as a model of ISR.
- Related to the previous point, how do ISR-related signatures correlate with some of the targets of this study, such as Cav1 and EGF themselves? Also, ISR activation might also explain, regardless of an indirect effect of Egr1, the regulation of the (apparently) key cytokine IL8, as this gene is also responsive to models for example of ER stress. What ISR branch is expected to be operating? The ample current range of genetic and pharmacological tools to specifically modulate the ISR network, should be at the very least explicitly discussed.
- Whether levels of Cavin1 do (or do not!) correlate with the gene signatures studied here would constitute an interesting piece of information. As the authors very briefly point out in their discussion, PTRF was initially proposed as a direct regulator of ribosomal biogenesis. In fact some of the effects the authors describe as consequence of Cav1 knockdown might well be related with the dynamics of this factor. Are there any changes in the levels and/or localisation of Cavin1 upon exposing their experimental models to ribosome poisoning? This is not trivial, because to the best of my knowledge, *C. elegans* does not have a true homolog of Cavin1 and does not form caveolae. This parallel comparison might shed interesting light on this question.
- Cav1 regulation by ISR (or, as solely shown in their study, by ribosomal poisoning), is it uniquely at the level of protein, or also at transcriptional level? Again, if the model of ribosomal poisons only affects Cav1 at protein level, this might hint at different molecular mechanisms as compared to real

ISR observed in IBD. Have the authors attempted to monitor the ratio of Cav1 isoforms? Again, this is not trivial as their compartmentalization is very different, and their segregation is also conserved in *C. elegans*.

- Does a nuclear-restricted EGFR construct bypass the regulatory mechanisms described here?

MINOR COMMENTS:

- I agree on the interpretation the authors aim to convey with figure 1G, but for the average reader in the caveolae field, I am afraid the images should be acquired with better resolution and image quality (avoid saturation). Quantitation will be demanded by experienced readers too.

- depletion of Cav1 must be assessed at protein level, not just transcript level (fig 1H, 4L...).

- While the structure of the text is good, I would encourage the assistance of a native English speaker for grammar and orthographical correction.

MS ID#: **COMMSBIO- 19-0894**

MS TITLE: **“Caveolar Communication with Xenobiotic-stalled Ribosome Deteriorates Gut Barrier Integrity”**

Dear Referees,

We really appreciate your kind comments on my manuscript. It was very nice chance to look deeply again into the results and scientific and translational meaning with your precious comments and we did best to come up with the every requirements from the reviewers. We will follow up each comment and suggest responses point-by-point to improve experimental results and resubmit the revised manuscript.

Sincerely yours,

Yuseok Moon, corresponding author

While both reviewers find this study interesting, they raised serious concerns about the assumptions of this study. As requested by these reviewers, we ask you to please show that ribosomes are inactivated in IBD patients and/or murine colitis and that ribosomal poisons indeed upregulate ATF4 or CHOP. We also ask that this manuscript goes through a professional copy-editing to help readers appreciate this study better.

Comments of Reviewer 1

In this manuscript, park S et al showed that in IBD ribosome inactivation triggered caveolar clustering and activation which counteracted EGFR mediated signaling pathway and gut epithelial protection. This may provide a new insight into the pathogenesis of IBD. However, the results presented in this manuscript are not convincing to draw this conclusion, and there are multiple questions/concerns need to be further addressed.

Major concerns:

1. The authors assumed that there presented ribosome inactivation in IBD, and did all the experiments based on this hypothesis. However, this manuscript did not provide evidence/results to show the presence of ribosome inactivation in IBD patients and/or murine colitis.

Response: As a reviewer pointed out, we performed additional experiments and analyses for this association in the clinical data and experiments (1E,F,G,H, 2B/C/D).

Patients with the ulcerative lesions were assessed for expression of EIF2AK2 (protein kinase R, PKR) as the ribosomal inactivation stress-responsive eIF2 \$\alpha\$ kinase. Moreover, PKR-linked biological responses were simulated in the experimental gut models of ribosome-inactivating stress.

In the clinical datasets: “Based on the assumption that the changes in Cav-1 as the signaling platform may reflect stress responses to chronic luminal insults in patients with IBD, four different global stress-related mammalian eIF2 α kinases, including EIF2AK1 (HRI), EIF2AK2 (PKR), EIF2AK3 (PERK), and EIF2AK4 (GCN2), were evaluated in both healthy and IBD patient groups based on a clinical genomic dataset (Fig. 1E). Among these, PKR was notably elevated in patients with UC and CD, including colon-only CD (cCD) and ileo-colonic CD (iCD), when compared to the control group. Moreover, disease-active patients displayed significantly enhanced PKR and Cav-1 levels compared to normal controls or disease-inactive IBD groups (Fig. 1F). In terms of signaling pathways, IBD patients with high PKR expression tended to show increased levels of PKR-activated stress-responsive transcription factors, such as activating transcription factor 4 (ATF4) and C/EBP homologous protein (CHOP) (Fig. 1G). Finally, the same clinical datasets demonstrated that IBD patients with high PKR expression also displayed enhanced Cav-1 expression (Fig. 1H), indicating a positive association between Cav-1 and PKR-linked stress responses.”

In the cellular levels: “PKR-linked stress responses were simulated in the experimental models of ribosomal stress via direct activation of PKR^{10, 19}. We assessed the effects of PKR activation on Cav-1 expression in the ribosomal stress-insulted murine gut and human cell models. Upon exposure to internal or external stressors, ribosomes stand sentinel. Stress-driven ribosomal stalling triggers eIF2 \$\alpha\$ -mediated global translational inhibition via PKR, which was mimicked by exposure to chemical ribosome-inactivating stressors (RIS). RIS-exposed mice displayed significantly elevated intestinal Cav-1 expression (Fig. 2A). Consistent with clinical transcriptomic analysis in patients with IBD (Fig. 1G), RIS-exposed cells showed initially increased levels of PKR-activated stress-responsive transcription factors, such as ATF3, ATF4, and CHOP, while expression of ER chaperone glucose-regulated protein 78 (GRP78) was suppressed (Fig. 2B). ----- Moreover, since ATF3 is a key transcription factor for stress-responsive genes during translational arrest²¹, we tested its effect on subsequent

Cav-1 expression and found that ATF3 was positively involved in inducing Cav-1 expression under ribosomal inactivation stress (Fig. 2D)."

2. The authors showed the gene expression level of Cav-1 in IBD. How about their protein expression levels? This is important to support the authors' hypothesis.

Response: As a reviewer pointed out, we assessed the protein levels using the proteomic analysis.

"Moreover, Cav-1 protein levels in colonic biopsies were higher in patients with IBD than in the healthy controls (Fig. 1B)."

3. To induce ribosome inactivation, the authors used RIS-1 in some experiments, and used RIS-2 in other experiments, and used both RIS-1 and RIS-2 in some experiments. What makes the authors to decide to use either RIS-1 or RIS-2? Was it due to inconsistent results caused by RIS-1 and RIS-2?

Response: Although there were differences in degree of actions, RIS-1 and RIS-2 showed similar patterns of regulation in tested models. *In particular, their nature of regulation on PKR-associated integrated stress responses (ISR) displayed similar trends. Both RIS-1- and RIS-2-exposed cells commonly showed early increased levels of PKR-activated stress-responsive transcription factors such as ATF3 (Activating Transcription Factor 3) and CHOP while expression of ER chaperone GRP78 (glucose-regulated protein 78) was suppressed by RIS-1 or RIS-2 (Fig. 2B). Since RIS-2 was a little more effective in enhancing Cav-1 expression than RIS-1 (Fig. 2B and 2C), RIS-2 was commonly tested in all other models (human cells, mouse, and C. elegans models).*

4. Previous publications showed that Cav-1 is not increased in the inflamed mucosa of patients with ulcerative colitis (Andoh, et al. Inflammatory Bowel Disease, 2001, 7(3):210-214), and Cav-1 was decreased in murine colitis and showed protective role for murine colitis (Weiss C, et al. Plos One, 2015). However, in the present study, the authors showed the increased Cav-1 expression in IBD patients. How will the authors explain the differences? It should be discussed in the discussion.

Response: As suggested, we discussed these points in the discussion section.

In the clinical data: As a reviewer pointed out, we addressed this issue by describing "*An opposing report on Cav-1 expression in patients*³² *performed non-statistical*

immunohistochemical analysis in 10 UC and 10 CD patients and observed a decreasing trend in Cav-1 expression. In contrast, the present analyses used two well-organized clinical datasets of patients with IBD (n=190 and n=194), resulting in statistically significant evaluations of Cav-1 expression (Fig. 1A). Moreover, clinical proteomic evaluation also supported the finding that Cav-1 is elevated in patients with IBD (Fig. 1B)”.

In the animal model data: As a reviewer pointed out, we addressed this issue by describing “*In contrast with the adverse activity of Cav-1 in the present model, a previous investigation using a 2,4,6-trinitrobenzenesulfonic acid (TNBS)-induced colitis model suggested that Cav-1 played protective roles against intestinal inflammation with no molecular evidence of its effects³³. TNBS-induced colitis is T cell-dependently regulated, leading to Cav-1 suppression³³. In contrast, we attempted to exclude the effects of acquired immunity and used a DSS-induced colitis model, which is mostly T cell-independent. Moreover, another study also demonstrated increased Cav-1 expression in a DSS-induced colitis model³⁴, which was consistent with the experimental results and clinical evaluation of datasets determined herein.*”

5. This manuscript is not well written. The purposes of some chemical treatment were not clearly mentioned. There were numerous English grammars found in this manuscript, and some part was hard to understand. Strongly suggest the authors to ask a Native English speaker to polish the English language of this manuscript.

Response: As a reviewer pointed out, we again got the more intensive language editing by native editor from the editing company (Job code PUSU_5333, *Editage* by Cactus).

6. The majority of in vitro assay was based on HCT-8 cell line. Did the authors try other human intestinal cell line and get similar findings?

Response: As a reviewer pointed out, other enterocyte-derived cell lines (HT-29 and SW-480) also showed enhanced expression of Cav-1 in response to ribosomal inactivation (**Fig. S1B**).

7. In Fig 3, the authors showed the epithelial chemokine production was decreased by EGFR inhibition, but increased the neutrophil and macrophage recruitment. What’s the reason to cause this? What’s the role of EGFR-induced chemokine?

Response: As a reviewer pointed out, it is a really great question. We addressed this issue by performing experiments (**Fig. 4G/H/I**) and described as “*these findings suggest that EGFR-induced chemokines are no longer positively involved in inflammatory cell recruitment during ribosomal inactivation stress. Instead, increased inflammatory cell recruitment in the EGFR-inhibited gut can be explained by other causes. Thus, it was hypothesized that EGFR-induced chemokines play protective roles against pro-inflammatory stimuli in the gut barrier. Ribosomal inactivation decreased the mucosal barrier against luminal microbiota, which was*

further aggravated by EGFR inhibition in both the small and large intestines of mice (Fig. 4G), indicating that EGFR had protective roles against bacterial exposure in the mucosal layer. Moreover, in vitro analyses of the permeability of the human epithelial cell monolayer demonstrated that IL-8 was positively involved in EGFR-mediated barrier protection from ribosomal inactivation stress (Fig. 4H and 4I). These findings suggest that EGFR-induced chemokines are essential to epithelial restitution rather than inflammatory cell recruitment during ribosomal inactivation. Therefore, barrier disruption in the EGFR-inhibited gut increased exposure to luminal microbes and subsequent inflammatory stress.”

Minor concerns:

1. Does the expression of Cav-1 correlate with the severity of inflammation?

Response: As a reviewer pointed out, we performed additional analysis and compared the Cav1 levels between disease active and inactive groups. *“disease-active patients displayed significantly enhanced PKR and Cav-1 levels compared to normal controls or disease-inactive IBD groups (Fig. 1F)”*.

2. Did you perform the correlation analysis on the expression levels of Cav-1 and EGFR? Although Fig 1E kindly supported that Cav-1 expression is negatively correlated with EGFR expression in IBD.

Response: As a reviewer pointed out, we performed additional analysis of correlation. *“Further analysis of the correlation between Cav-1 and EGFR levels confirmed that EGFR expression was inversely related with Cav-1 levels in mucosal biopsies from patients with IBD (Fig. S1C).”*

3. In Figure legends it missed the repeating number of experiments performed or number used.

Response: As a reviewer pointed out, all figure legends were inspected for this issue and added description for number of experiments of in vitro evaluation and animal numbers.

4. Figure legends were not well prepared.

Response: As a reviewer pointed out, we checked the all legends in the manuscript by authors and native speakers.

5. Some figures miss scale bar. eg, Fig 3G

Response: As a reviewer pointed out, missing bars were added in all figures.

6. Fig 5G, the statistical significance among groups is not clear.

Response: As a reviewer pointed out, it was fixed as “*Different letters over each bar represent significant differences between groups in each compartment (the whole cell, cytoplasm, or nucleus) (Fig. 6N)*”

Comments of Reviewer 2

Understanding the cellular and molecular principles by which tissues are organized, to ensure their integrity and repair, is essential to understand disease. In this report, Park and colleagues contribute insights onto how general cell stress mechanisms (the Integrated Stress Response) rewires the compartmentalization of the essential Egfr signalling network, in a manner dependent on caveolae, to determine both epithelium activation and repair and the establishment of inflammation.

Conceptually, their study is interesting, in part because of the effort they make to overarch and integrate several mechanisms among each other. While it is not explicitly stressed in the report, they actually build their hypothesis from patient-derived gene expression datasets. I personally consider many ideas they propose of interest to a broad readership. There are however a number of points that the authors should address and discuss, mostly pertaining to their findings on Cav1 biology and ISR regulation.

MAJOR COMMENTS:

1. A first major concern is the assumption that pharmacological stalling of polysomes should faithfully reproduce the events and dynamics downstream the activation of any branch of the ISR, which involves dysassembly of polysomes, potentially in a rather compartmentalized

manner. Do they observe downstream hallmarks of ISR activation, such as upregulation of ATF4 or CHOP? I can understand the aim of overarching all those different "modalities" of ISR activation, but the authors should at the very least discuss the comparison of both conditions, and provide more extensive literature on the use of these ribosomal poisons as a model of ISR.

Response: As a reviewer pointed out, we performed additional experiments and analyses for this association in the clinical data and experiments (1E,F,G,H, 2B/C/D).

Patients with the ulcerative lesions were assessed for expression of EIF2AK2 (protein kinase R, PKR) as the ribosomal inactivation stress-responsive eIF2 \$\alpha\$ kinase. Moreover, PKR-linked biological responses were simulated in the experimental gut models of ribosome-inactivating stress.

In the clinical datasets: *“Based on the assumption that the changes in Cav-1 as the signaling platform may reflect stress responses to chronic luminal insults in patients with IBD, four different global stress-related mammalian eIF2 α kinases, including EIF2AK1 (HRI), EIF2AK2 (PKR), EIF2AK3 (PERK), and EIF2AK4 (GCN2), were evaluated in both healthy and IBD patient groups based on a clinical genomic dataset (Fig. 1E). Among these, PKR was notably elevated in patients with UC and CD, including colon-only CD (cCD) and ileo-colonic CD (iCD), when compared to the control group. Moreover, disease-active patients displayed significantly enhanced PKR and Cav-1 levels compared to normal controls or disease-inactive IBD groups (Fig. 1F). In terms of signaling pathways, IBD patients with high PKR expression tended to show increased levels of PKR-activated stress-responsive transcription factors, such as activating transcription factor 4 (ATF4) and C/EBP homologous protein (CHOP) (Fig. 1G). Finally, the same clinical datasets demonstrated that IBD patients with high PKR expression also displayed enhanced Cav-1 expression (Fig. 1H), indicating a positive association between Cav-1 and PKR-linked stress responses.”*

In the cellular levels: *“PKR-linked stress responses were simulated in the experimental models of ribosomal stress via direct activation of PKR^{10, 19}. We assessed the effects of PKR activation on Cav-1 expression in the ribosomal stress-insulted murine gut and human cell models. Upon exposure to internal or external stressors, ribosomes stand sentinel. Stress-driven ribosomal stalling triggers eIF2 α -mediated global translational inhibition via PKR, which was mimicked by exposure to chemical ribosome-inactivating stressors (RIS). RIS-exposed mice displayed significantly elevated intestinal Cav-1 expression (Fig. 2A). Consistent with clinical transcriptomic analysis in patients with IBD (Fig. 1G), RIS-exposed cells showed initially increased levels of PKR-activated stress-responsive transcription factors, such as ATF3, ATF4, and CHOP, while expression of ER chaperone glucose-regulated protein 78 (GRP78) was suppressed (Fig. 2B). --- Moreover, since ATF3 is a key transcription factor for stress-responsive genes during translational arrest²¹, we tested its effect on subsequent Cav-1 expression and found that ATF3 was positively involved in inducing Cav-1 expression under ribosomal inactivation stress (Fig. 2D).”*

2. Related to the previous point, how do ISR-related signatures correlate with some of the targets of this study, such as Cav1 and EGF themselves? Also, ISR activation might also explain, regardless of an indirect effect of Egr1, the regulation of the (apparently) key cytokine IL8, as this gene is also responsive to models for example of ER stress. What ISR branch is expected to be operating? The ample current range of genetic and pharmacological tools to specifically modulate the ISR network, should be at the very least explicitly discussed.

Response: It is a great comment. As a reviewer pointed out, we performed additional analyses.

- IBD patients with high expression of PKR displayed enhanced levels of Cav-1 expression (**Fig. 1H**), indicating the positive association between Cav-1 and PKR-linked stress responses.
- In terms of ISR, high-PKR patients showed attenuated levels of EGFR expression in intestine (**Fig. 2H**), indicating a potent negative regulation of EGFR during PKR activation.
- Mechanistically, we tried to find the **ISR branch** for Cav-1 and EGFR regulation by performing additional experiments (**Fig. 2D, 5G/H**).

① **ISR-ATF3-CAV1:** RIS-exposed cells showed initially increased levels of PKR-activated stress-responsive transcription factors, such as ATF3, ATF4, and CHOP, while expression of ER chaperone glucose-regulated protein 78 (GRP78) was suppressed (Fig. 2B). Moreover, caveolar components such as cavin-1 and Cav-1 displayed upregulated expression. Compared with early induction of ATF3, CHOP, and cavin-1, late induction was observed for Cav-1, including alpha and beta isoforms, in response to ribosomal inactivation (Fig. 2B and 2C). The alpha isoform was more responsive than beta isoforms to ribosomal stress. Moreover, since ATF3 is a key transcription factor for stress-responsive genes during translational arrest²¹, we tested its effect on subsequent Cav-1 expression and found that ATF3 was positively involved in inducing Cav-1 expression under ribosomal inactivation stress (Fig. 2D).

② **ISR-PKR-p38/EGFR pathway:** In addition to the involvement of EGFR-mediated signaling in chemokine production, PKR, the ribosomal stress-responsive eIF2 α kinase, was tested for its effects on p38 MAPK signaling and chemokine expression. Subsequently, shRNA-mediated PKR suppression attenuated p38 activation (Fig. 5G) and chemokine induction by RIS (Fig. 5H). Moreover, PKR was involved in activating EGFR phosphorylation, leading to p38 MAPK activation (Fig. 5G). Altogether, these findings indicated that PKR is a positive upstream regulator of EGFR and p38 MAPK signaling and subsequent chemokine production under ribosomal inactivation stress in human intestinal

epithelial cells.

3. Whether levels of Cavin1 do (or do not!) correlate with the gene signatures studied here would constitute an interesting piece of information. As the authors very briefly point out in their discussion, PTRF was initially proposed as a direct regulator of ribosomal biogenesis. In fact some of the effects the authors describe as consequence of Cav1 knockdown might well be related with the dynamics of this factor. Are there any changes in the levels and/or localisation of Cavin1 upon exposing their experimental models to ribosome poisoning? This is not trivial, because to the best of my knowledge, *C. elegans* does not have a true homolog of Cavin1 and does not form caveolae. This parallel comparison might shed interesting light on this question.

Response: As a reviewer pointed out, I addressed this issue by performing additional analysis and experiments (Fig. 2B).

Correlation analysis: *In contrast with Cav-1, cavin-1 was not correlated with EGFR expression although cavin-1 was elevated in patients with IBD (unpublished data, depicted in the discussion section).*

Cellular model: Caveolar components such as cavin-1 and Cav-1 displayed upregulated expression in response to ribosome inactivation. “Compared with early induction of *ATF3*, *CHOP*, and *cavin-1*, late induction was observed for *Cav-1*, including alpha and beta isoforms, in response to ribosomal inactivation (Fig. 2B and 2C). Moreover, since *ATF3* is a key transcription factor for stress-responsive genes during translational arrest²¹, we tested *ATF3* effect on subsequent *Cav-1* expression and found that *ATF3* was positively involved in inducing *Cav-1* expression under ribosomal inactivation stress (Fig. 2D).”

4. Cav1 regulation by ISR (or, as solely shown in their study, by ribosomal poisoning), is it uniquely at the level of protein, or also at transcriptional level? Again, if the model of ribosomal

poisons only affects Cav1 at protein level, this might hint at different molecular mechanisms as compared to real ISR observed in IBD. Have the authors attempted to monitor the ratio of Cav1 isoforms? Again, this is not trivial as their compartmentalization is very different, and their segregation is also conserved in *C. elegans*.

Response: As a reviewer pointed out, we performed additional experiment using different sets of primers (**Fig. 2C**). Since it was not clear to address the isoforms at the protein levels, we quantified the transcript levels. *“Compared with early induction of ATF3, CHOP, and cavin-1, late induction was observed for Cav-1, including alpha and beta isoforms, in response to ribosomal inactivation (Fig. 2B and 2C). The alpha isoform was more responsive than beta isoforms to ribosomal stress.”*

5. Does a nuclear-restricted EGFR construct bypass the regulatory mechanisms described here?

Response: It is also a great question. *Cav-1-proteasome pathway mediated the degradation of EGFR protein, which also can occur in the nucleus since the proteasome machinery is indeed working in the nuclear compartment*³¹(Discussion section). Regardless of the proteasomal regulation of EGFR protein, ISR-responsive PKR activates p38 MAPK signaling (**Fig. 5G**) which is crucial for EGFR entry into nuclei for transcriptional regulation (**Fig. 6H/N**). Therefore, while cellular levels of EGFR is regulated by the proteasomal machinery, the cellular nuclear translocation and transcriptional actions of EGFR are affected by PKR-p38 MAPK signaling under the stress of ribosomal inactivation.

We depicted it in the discussion section as *“Cav-1-mediated EGFR degradation can occur in the nucleus because the proteasome machinery functions also in the nuclear compartment*³¹. *In spite of proteasomal actions, ISR-responsive PKR activated p38, which was crucial for the cellular translocation and transcriptional actions of EGFR under ribosomal inactivation stress.”*

MINOR COMMENTS:

1. I agree on the interpretation the authors aim to convey with figure 1G, but for the average reader in the caveolae field, I am afraid the images should be acquired with better resolution and image quality (avoid saturation). Quantitation will be demanded by experienced readers too.

Response: As a reviewer pointed out, we added the quantitative analysis of caveolae puncta in **figure 2E (boxed graph)**.

2. depletion of Cav1 must be assessed at protein level, not just transcript level (fig 1H, 4L...).

Response: As a reviewer pointed out, we added Cav-1 protein levels (Fig. 2I, 6F).

3. While the structure of the text is good, I would encourage the assistance of a native English speaker for grammar and orthographical correction.

Response: As a reviewer pointed out, we again got the more language editing by a native editor from the editing company (Job code PUSU_5333, *Editage* by Cactus).

REVIEWERS' COMMENTS:

Reviewer #1 (Remarks to the Author):

The authors have addressed the majority of my concerns. Although the authors replied my comment about RIS-1 vs RIS-2 (see comment #3), the authors should clearly explain this in the manuscript, so the readers can understand.

Reviewer #3 (Remarks to the Author):

A number of novel items (mostly fig. 2B) partly address some of my previous remarks.

In terms of mechanistic interpretation of the observations reported, a missing refinement is to specifically trigger PKR activity with a 'natural' stimulus (taken together, and assuming the model proposed, polyI:C transfection might recapitulate the effects exerted by pharmacological ribosome stalling). While experimentally this approach is very challenging (and I understand these are difficult times for experimentation...) I would firmly encourage the authors to discuss and propose the source of this stress response in vivo.

Otherwise and beyond standard proofreading, I deem the manuscript suitable for publication.

MS ID#: **COMMSBIO- 19-0894B**

MS TITLE: **“Caveolar Communication with Xenobiotic-stalled Ribosomes Compromises Gut Barrier Integrity”**

Dear Referees,

We really appreciate your kind comments on my manuscript. It was very nice chance to look deeply again into the results and scientific and translational meaning with your precious comments and we did best to come up with every requirement from the reviewers. We will follow up each comment and suggest responses point-by-point to improve experimental results and resubmit the revised manuscript.

Sincerely yours,

Yuseok Moon, corresponding author

Comments of Reviewer 1

Remarks to the Author:

The authors have addressed the majority of my concerns. Although the authors replied my comment about RIS-1 vs RIS-2 (see comment #3), the authors should clearly explain this in the manuscript, so the readers can understand.

Response: As a reviewer pointed out, we added the description as *“Although there were differences in their degrees of action, RIS-1 and RIS-2 showed similar patterns of gene regulation for PKR-associated integrated stress responses (ISR). However, since RIS-2 had more remarkable effects on Cav-1 induction than RIS-1 (Fig. 2C and 2D), it was used as a representative modulator of caveolin-linked signaling under ribosomal inactivation stress in this study.”*

Comments of Reviewer 2

A number of novel items (mostly fig. 2B) partly address some of my previous remarks. In terms of mechanistic interpretation of the observations reported, a missing refinement is to specifically trigger PKR activity with a 'natural' stimulus (taken together, and assuming the model proposed, polyI:C transfection might recapitulate the effects exerted by pharmacological ribosome stalling). While experimentally this approach is very challenging (and I understand these are difficult times for experimentation...) I would firmly encourage the authors to discuss and propose the source of this stress response in vivo.

Response: As a reviewer pointed out, this comment is very excellent. We described details on PKR and their clinical implications and proposals for better understanding in the diseases as *“Clinical dataset-based analysis in patients with IBD indicated a remarkably elevated expression of PKR and stress responsive genes in a PKR-dependent manner. PKR-linked biological responses were simulated in experimental gut models of ribosome-inactivating stress. However, mammalian PKR is activated mainly by double-stranded RNA (dsRNA) produced during viral infection^{62, 63, 64}. In addition to dsRNA, single-stranded RNA also binds to PKR and causes a conformational change, leading to its dimerization via its C-terminal kinase domain and subsequent autophosphorylation during viral infection^{63, 65}. Activated PKR inhibits viral and host protein synthesis through eIF2 α phosphorylation, which facilitates the stress responses and antiviral defense. Moreover, PKR can be activated by other diverse stresses, such as oxidative and endoplasmic reticulum stress, metabolic stress, or by other chronic inflammatory stresses in a dsRNA-independent manner^{12, 66, 67, 68}. Therefore, although we focused on a ribosomal stress-based assessment of PKR-linked events, additional evaluations are warranted in pathologic states, including viral infection, tumorigenesis, and diet-associated metabolic stress. Such extensive assessment would explain the complex PKR-linked translational network in the patients experiencing gut distress throughout their lives.”*